

# Continental-scale bias-corrected climate and hydrological projections for Australia

Justin Peter[1,5], Elisabeth Vogel[1,3], Wendy Sharples[1], Ulrike Bende-Michl[1], Louise Wilson[2], Pandora Hope[1], Andrew Dowdy[1], Greg Kociuba[1], Sri Srikanthan[1], Vi Co Duong[1], Jake Roussis[1], Vjekoslav Matic[1], Zaved Khan[1], Alison Oke[1], Margot Turner[1], Stuart Baron-Hay[1], Fiona Johnson[3], Raj Mehrotra[3], Ashish Sharma[3], Marcus Thatcher[4], Ali Azarvinand[1], Steven Thomas[1], Ghyslaine Boschat[1], Chantal Donnelly[1] and Robert Argent[1]

[1]Bureau of Meteorology, Melbourne, Victoria, Australia
[2]Met Office, International Climate Services, Exeter, United Kingdom
[3]Water Research Centre, School of Civil and Environmental Engineering, The University of New South Wales, Sydney, New South Wales, Australia
[4]CSIRO Marine and Atmospheric Research, Aspendale, Victoria, Australia
[5]Centre for Applied Climate Sciences, University of Southern Queensland, Toowoomba, Queensland, Australia

*Correspondence to*: Ulrike Bende-Michl (ulrike.bende-michl@bom.gov.au)

**Abstract.** The Australian Bureau of Meteorology has developed a national hydrological projections (NHP) service for Australia. With the focus on hydrological change assessment, the NHP service aims at being complementary to climate projections work carried out by many federal and state governments, universities, and other organisations across Australia. The projections comprise an ensemble of application-ready bias-corrected climate model data and derived hydrological projections at daily temporal and $0.05° × 0.05°$ spatial resolution for the period 1960–2099 and two emission scenarios (RCP 4.5 and RCP 8.5). The spatial resolution of the projections matches that of gridded historical reference data used to perform the bias correction and the Bureau's operational gridded hydrological model. Three bias correction techniques were applied to four CMIP5 global climate models (GCMs) and one to output from a regional climate model forced by the same four GCMs, resulting in a 16-member ensemble of bias-corrected GCM data for each emission scenario. The bias correction was applied to fields of precipitation, minimum and maximum temperature, downwelling shortwave radiation and surface winds. These variables are required inputs to the Bureau's landscape water balance hydrological model (AWRA-L) which was forced using the bias-corrected GCM and RCM data to produce a 16-member ensemble of hydrological output. The hydrological output variables include root-zone soil moisture (moisture in the top 1 m soil layer), potential evapotranspiration and runoff. Here we present an overview of the production of the hydrological projections, including GCM selection, bias correction methods and their evaluation, technical aspects of their implementation and examples of analysis performed to construct the NHP service. The data are publicly available on the National Computing Infrastructure (https://dx.doi.org/10.25914/6130680dc5a51) and a user interface is accessible at https://awo.bom.gov.au/products/projection/.



## 1. Introduction

Australia's climate has a large natural variability encompassing many different climate zones and multiple climate drivers leading to frequent floods and extended periods of droughts. In addition, Australia's climate is changing with increasing temperatures and shifts to rainfall patterns impacting all components of the hydrologic cycle. Seasonal precipitation has changed across the country, broadly increasing in the north and decreasing in the south (Dey et al., 2019) with similar patterns evident in streamflow (Zhang et al., 2016). The *State of the Climate 2022* (CSIRO and Bureau of Meteorology, 2022) reports that, in Australia's southwest, cool-season (April–October) precipitation has declined by around 15% since 1970. Across the same region, May to July rainfall has seen the largest decrease, by around 19 per cent since 1970. In the south-east of Australia, precipitation started to decline around 1990, and the average cool-season precipitation from 2000 to 2019 was 10% less than last century. Along with this observed decline in precipitation, streamflow has declined substantially in both the south-west and south-east; changes in streamflow are typically disproportionly larger than changes in precipitation (Chiew, 2006; Wasko et al., 2021; Zhang et al., 2016).

To deal with changes to water availability, Australia's water policy and infrastructure investment decisions require high-resolution hydroclimate information representative of past, current, and future variability. Of prime interest are precipitation changes, and how those changes will impact runoff, soil moisture and evapotranspiration across Australia. At present, many water utilities, planners and operators use in-house hydrological models to manage their resources at strategic timeframes of 20–50 years (or more), whereby the simulations of streamflow feed into decision-making, planning and implementation which often requires significant investment. Traditionally the modelling approach for these studies has been to use the historic climate record of a variable (e.g. rainfall, runoff) for a region or catchment of interest and extend it using stochastic methods to increase the sample of extreme events (Crosbie et al., 2009; CSIRO, 2009). However, this approach is insufficient to adequately represent the range of plausible climate futures. Several Australian states have commissioned and produced downscaled climate projections of rainfall and in some cases runoff. However, a key issue is that the methodologies, ensemble sizes, emissions scenarios, and sets of host GCMs used, varies considerably so the data are too heterogenous for use in applications that intersect jurisdictional boundaries. Often a clear description of downscaling, bias correction and uncertainty analysis is not provided.

Detailed climate projections for Australia have been produced by the Bureau and the Commonwealth Scientific and Industrial Research Organisation (CSIRO) and resulted in the Climate Change In Australia service (CSIRO and Bureau of Meteorology 2015; hereon CCiA). The CCiA service provides application-ready data and information on climate projections for Australia, using a subset of eight GCMs, selected from all CMIP5 GCMs for their ability to reproduce climate features that are important for the Australian climate and suitable for use in impact models. The application-ready data was produced using a mean scaling for all variables other than precipitation for which quantile-quantile scaling was applied. However, the application-ready data





is not transient[1] and explicit hydrological output was not provided. Data currently available for runoff projections that are sufficient to cover multiple catchments from one data source has been developed by Chiew et al. (2018) (see also, Zheng et al., 2019b). This is an ensemble of national projections of runoff using a delta-change methodology that applies the mid-century change signal from 42 GCMs to historical time series which are then run through GR4J  rainfall-runoff models (Perrin

et al., 2003)for each catchment. As with any scaled dataset, applications are limited to uses for which transient series are not required and assume the variability and persistence in the observed period (for example the daily distribution and sequencing of precipitation) is representative of a future climate undergoing warming. Ensembles of daily hydrological projections produced from bias-corrected climate model output are available globally, including Australia, via ISIMIP[2] at a spatial resolution of 0.5 deg (approx. 50km). These application of this data is limited by the relatively coarse spatial resolution, and

there is no information describing the suitability of the bias-correction reference data for Australian conditions.

To address these deficits in hydrological projections the Australian Bureau of Meteorology (the Bureau) has produced an ensemble of daily projections of hydrological variables at $0.05° \times 0.05°$ (approximately 5 km) spatial resolution spanning the period 1960–2099. In addition, application-ready data sets of the variables required to produce the hydrological output at the

same temporal and spatial resolution are also available. These were produced as a central component of the Bureau's National Hydrological Projections (NHP) project. The overarching goal of the NHP was to provide a nationally consistent set of projections, including publicly accessible data sets as well as communication tools that will be useful to water-sensitive sectors of the Australian community to prepare for, and adapt to, the many possible impacts of climate change, especially those sectors for which the hydrological cycle is of importance. By developing these data sets we aim at addressing these limitations and

providing nationally consistent hydrological projections for the Australian community.

Development of the application-ready climate data and the derived hydrological projections required several considerations:

- GCM selection and the associated choice of emission scenarios,
- A method to downscale the GCM output to the spatial scale required for the hydrological model, including techniques
to correct for the biases of the GCM,
- An observational data set, which is required for both implementation and evaluation of bias correction,
- Choice of a hydrological model to run using the downscaled climate data, and,
- Development of communication tools for users of the projections data.

---

[1] Transient projections are those which are applicable for the entire duration of the projection period. The scaling methodology used in CCiA requires that the correction be applied applied in discrete time chunks (30 years) and therefore not suitable to be used as a continuous timeseries.

[2] https://esg.pik-potsdam.de/search/isimip/

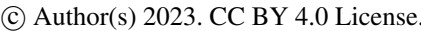
Each of these decisions was guided by both pragmatic and scientific considerations. The various components of these decisions and the resultant modelling chain are shown in Figure 1. The objective of this paper is to detail the decisions made and methodologies employed to develop a national continental-scale hydrological projections service.

The structure of the paper broadly follows the dot points above: Section 2 details the GCM selection process, including how the selected GCMs compare to the larger collection of CMIP5 models, and the model used to dynamically downscale the GCMs. The reference observational data and input GCM scale is described in Section 3. The bias correction methods implemented are reviewed in Section 4. The AWRA-L hydrological model is described in Section 5 and a brief evaluation of the bias correction methods against the reference observational data in Section 6. The GCM and hydrological projections and

some example analyses are detailed in Section 7, with a focus on extreme rainfall and runoff events in Section 7.4. We provide details of how the data sets were used to provide communication of the projections for end users in Section 8. Finally, conclusions, considerations around uncertainty in the projections and details of code availability are presented.

## 2. GCM Selection

The choice of the input GCM was guided by both pragmatic and scientific considerations. The AWRA-L hydrological model (see Section 5) requires daily values of precipitation (pr), maximum and minimum temperature (tasmax, tasmin), surface wind speed (sfcWind) and downwelling shortwave radiation (rsds) as inputs. Furthermore, the Australian research community uses CMIP5 GCM data stored on the National Computing Infrastructure (NCI) supercomputer, so the ready availability of these variables (at daily temporal resolution) was of paramount practical importance. Ideally, an exhaustive set of GCM data, for

instance, the complete set of 47 CMIP5 simulations assessed in CCiA would have been utilised, however, that was not feasible, requiring us to choose a subset. The CCiA report guided that decision making, wherein it recommended the use of 8 GCMs to be used to prepare application-ready data for Australia. They were chosen primarily on their ability to represent the climate drivers of significant importance for Australia, in particular, ENSO, the Southern Annular Mode (SAM) and monsoonal rainfall patterns, as well as to provide a reasonable representation of the general range of variation between CMIP5 models (CSIRO

and Bureau of Meteorology, 2015). Using this guidance and the practical considerations around data availability, the final suite of models chosen was ACCESS1-0, CNRM-CM5, GFDL-ESM2M and MIROC5 (see Table 1).

Adjunct choices around emission scenarios were also required; simulations produced using the RCP4.5 and RCP8.5 emission pathways were chosen. These emissions pathways were chosen to provide a high (RCP8.5) and moderate (RCP4.5) set of

temperature projections, noting that the set of modelled greenhouse gas emission pathways provided in CMIP5 have relatively minimal deviation before 2050 and the observed climate change trends for $CO_2$ emissions and temperature in recent decades





indicate that the high emissions pathway (RCP 8.5) has been followed more closely than other emissions pathways (e.g., RCP 2.6) (Schwalm et al., 2020; Stocker et al., 2013).

## 2.1. Regional climate model (CCAM)

To extend the GCM data, the Conformal Cubic Atmospheric Model (CCAM; McGregor, 2005; McGregor and Dix, 2008) was
used to dynamically downscale the four GCMs. CCAM outputs contribute to the CORDEX (Coordinated Regional Downscaling Experiment)[3] international intercomparison of downscaling models. It is a global atmosphere-only climate model that has the ability for its grid to be stretched such that it has a finer resolution over the domain of interest (Australia) and coarser resolution in more distant regions (McGregor, 2005). In the configuration used, the mean and variance of the sea surface temperatures (SSTs) of the host GCM are bias corrected to provide boundary conditions for CCAM to produce 50 km
resolution projections of the atmospheric state over the Australian continent (Clarke et al., 2019; Hoffmann et al., 2016). As for the host GCMs, the simulations were run for the period 1960 to 2100, for both RCP4.5 and RCP8.5 scenarios. The resulting 50 km simulations represent a reconstruction of the atmosphere for which the GCM SSTs biases have been removed and the atmosphere is allowed to evolve independently of the host GCM.

The only information incorporated into CCAM from the host GCM is the bias-corrected SSTs, so the resultant atmospheric evolution may depart quite markedly from the host GCM. This has the effect that the range of the projections, and by implication the uncertainty, will differ from those that only include the host GCM. There are systematic differences between the GCM and CCAM projections, which we will explore in Section 7, however, the general effect of the inclusion of the CCAM simulations in the NHP ensemble is to increase the ensemble range. The approach to forcing the downscaling model
is the same as used for the Victorian Climate Projections – 2019 (Clarke et al., 2019) but differs from that used in some other projects, e.g. the Energy Sector Climate Initiative (ESCI), where a spectral nudging approach was used, in which case, the underlying signal from the GCMs will be more prominent.

## 2.2. Comparison to the CMIP5 ensemble

**It is informative to understand how our selection of GCMs fit within the larger ensemble of CMIP5 model simulations.**

Figure 2 and Figure 3 show the ranking of 40 GCMs used in CCiA, for surface temperature (tas) and precipitation (pr). The ranking was evaluated by calculating the areal (in this case all Australia) and temporal averaged difference of the projection period and the same quantity for the historical period; we will refer to as a "change factor". To examine the change factor, we define four future 30-year time slices: 2016–2045, 2036–2065, 2056–2085 and 2070–2099, which are centred on 2030, 2050,

---

[3] Further details about CCAM physics and versions can be found at https://confluence.csiro.au/display/CCAM/CCAM, including a list of relevant publications at https://confluence.csiro.au/display/CCAM/Key+publications+for+CCAM.



2070 and 2085, respectively, while the historical period was defined as 1976–2005. Due to Australia's large size and its geographical location, the climate of Australia varies markedly from the tropical north to temperate south hence nationally averaged precipitation may not provide meaningful insight from a climatological perspective, nevertheless, it does impart interpretation of how our choice of GCMs occupy the phase space spanned by the CMIP5 models. For surface temperature,

our selection falls approximately midway the range of the CCiA ensemble, however, still span a relatively large range of change factors. The spread amongst the change factor values generally increases as a function of time and is also slightly larger for the RCP8.5 scenario. The ACCESS1-0 and GFDL-ESM2M models have an increased warming signal compared to MIROC5 and CNRM-CM5. The change factor for precipitation consists of both wetting and drying signals across each time slice and scenario, although generally, two models (ACCESS1-0 and GFDL-ESM2M) show a drying signal and two show

increased precipitation (CNRM-CM5 and MIROC5). GFDL-ESM2M has a consistent and large drying signal, while the precipitation change signal from the ACCESS1-0 model varies between time slices and scenarios; CNRM-CM5 and MIROC5 both indicate enhanced precipitation.

Although focussing on a particular model is useful, when using the output from several models the range of uncertainty

(provided by the differing simulations) can be examined via constructing the model ensemble and calculating ensemble statistics. As for
Figure 2, the change factor is calculated for each model separately which are then aggregated to calculate ensemble statistics including ensemble median and the ensemble 10th and 90th percentiles. Here we introduce "change factor plots" an example of which is shown in Figure 4. It displays the model ensemble change factor spread for surface temperature (left) and precipitation

(right) for the NHP ensemble and the CCiA ensemble, averaged over Australia, for both RCPs and for the four future time slices. The NHP ensemble median is cooler than that of the CCiA ensemble and has a decreased ensemble range. The precipitation signals between the NHP ensemble and CCiA ensemble are often of opposite sign, especially for RCP4.5. Although the ensemble range is generally smaller for the NHP models, it is comparable to that of CCiA and in the case of the 2030 and 2050 time slices for RCP4.5, is larger.

Figure 4 indicates that the NHP ensemble provides a reasonable subsample of the CCiA ensemble. To focus on how the climate in a particular region is projected to change in the future it is useful to divide a large geographical region, such as Australia, into sub-regions. In keeping with the recommendations of CCiA, the Natural Resource Management (NRM) areas are used in this report, and we will refer to changes across NRM clusters (see Figure 5). These regions were chosen to reflect locally

similar climatic conditions, biophysical factors and broad patterns of climate change (CSIRO and Bureau of Meteorology, 2015). CCiA also included further sub-divisions that may be better suited for impact and assessment studies. However, our intention here is not to provide such assessments but rather a broad overview of the data available, including the bias correction methods, so that it can be used for such studies in the future.


Change plots were also constructed for the NRM clusters (not shown) and the discrepancy between the NHP and CCiA ensembles changes depending on the geographical region. For instance, the NHP warming signal in the Southern and Southwestern Flatlands (SSWF) NRM is not as large as the CCiA ensemble; however, the precipitation change signal for both the CCiA and NHP ensembles indicate a consistent drying trend although the NHP ensemble drying signal is less. As for all

of Australia, the range of temperature and precipitation projections of the NHP ensemble was evaluated against the 40 member CCiA ensemble and it was found that the NHP ensemble spanned the complete ensemble range remarkably well. This means that the four GCMs used in NHP include both cooler/warmer and dryer/wetter projections, giving us confidence that the mean changes of the NHP ensemble reflect that of the complete ensemble. Nevertheless, the spread of the NHP ensemble is less than that of CCiA, suggesting that the uncertainties are underestimated. However, we note that this analysis only considers the four

host GCMs and does not include CCAM simulations. Since CCAM develops a climate independent of the host GCM it has the effect of increasing the ensemble spread compared to that of the host GCMs. Although the inclusion of the CCAM ensemble increases the uncertainty of the projections, rather than considering this a limitation, it is advantageous as it enables sampling more of the phase space represented by the full suite of CMIP5 models (see

Figure 2, Figure 3 and Figure 4), thereby decreasing the reliance of the interpretation on the results on the choice of GCM

selection.

## 3.   Reference data

The output from GCMs and RCMs have systematic biases that need to be corrected before using their data for climate change impacts and risk assessment studies. This is achieved using bias correction methods which are designed to preserve certain

moments between the observations and the simulations. Implementation of a bias correction method requires three main components: (1) GCM simulations which are to be bias-corrected, (2) reference data, which can be direct observations or a derived product (e.g., gridded observations or reanalyses) and 3) a bias correction algorithm. The reference data sets required are those necessary to run the AWAR-L hydrological model (see Sect. 5) are precipitation (pr), maximum and minimum temperature (tasmax/tasmin), downwelling shortwave radiation (rsds) and surface wind speed (sfcWind). Here we describe

the reference data sets used to calibrate the GCM output over the historical period (defined to be 1976–2005).

### 3.1.  Precipitation and Temperature

Daily gridded precipitation and temperature data were obtained from the Australian Water Availability Project (AWAP) climate data set which consists of surface air temperature (daily minimum and maximum) and daily precipitation from 1 January 1911 to the present (Jones et al., 2009). The rainfall and temperature data are interpolated from daily station records

using a Barnes successive-correction scheme that applies weighted averaging to the station data and then provided on a 0.05° (approximately 5 km) spatial grid across Australia. Topographical information is included by using anomalies from long-term



(monthly) averages in the analysis process. For temperature, the daily anomaly is represented by the difference from the monthly mean, while for rainfall, the ratio of the daily observed to the monthly average precipitation is used.

Precipitation data represent the total recorded over the 24 hours beginning (and ending) at 09:00 local time (LT) and is assigned
to the day from which the recording began. The highest temperature over the 24 hours before the observation at 9 am is recorded as the maximum temperature ($Tmax$) for the previous day. The lowest temperature ($Tmin$) for the 24 hours before 9 am is recorded for the day on which the observation was made. This convention usually guarantees that $Tmin$ and $Tmax$ will have occurred on the same calendar day, however, in some instances it is possible that $Tmin > Tmax$. For the ISIMIP2b method (see Sect 4.14.1) the surface temperature (*tas*) is corrected (not $Tmin$ and $Tmax$) and the minimum and maximum are
derived (see Eq. 26 in Hempel et al., (2013)). As such, for the ISIMIP2b method, *tas* was derived from the AWAP data using their average (i.e. $tas = (Tmin + Tmax)/2$ ). In some circumstances tasmin can exceed tasmax ($Tmin > Tmax$), for which we set $Tmin = Tmax$. The other bias correction methods (MRNBC and QME) correct $Tmin$ and $Tmax$ directly, therefore no calculation of *tas* was required for these methods. Note that, while observations are made from 9 am to 9 am, the bias corrections are calculated by calendar day.

Since AWAP is a gridded data set it has errors compared to measurements from the original station data. This is particularly true in data-sparse areas for rainfall since it is highly variable both spatially and temporally. For this reason, some regions, particularly in central Western Australia have a mask applied and are excluded from the analysis. Furthermore, in areas of steep topography, particularly along the Great Diving Range, AWAP has been shown to underestimate rainfall (Chubb et al.,
2016). It has also been shown to underestimate extreme rainfall percentiles when compared to station data (King et al., 2013).

### 3.2.  Solar radiation and wind speed

Daily surface shortwave downwelling solar radiation flux (rsds) is obtained from geostationary satellites (Grant et al., 2008) and aggregated to the same 0.05° AWAP grid. The solar radiation record is from 1990 to the end of 2005 (the calibration period). From 1 January 1990, the GMS-4, GMS-5, GOES-9 and MTSAT-1R satellites were used, although these were
replaced by the Himawari-8 satellite from 23 March 2016. Daily climatological averages (for each day of the year) are used for solar radiation before 1990.

Site-based daily near-surface wind speed observations (sfcWind) collated by the Bureau of Meteorology and interpolated nationally (McVicar et al., 2008) were used for daily average wind speed from 1975 onwards (when sufficient site observations
were being collated by the Bureau) for the GCM bias correction. The near-surface wind data set is for the wind speed recorded at a height of 2 metres, however, since the GCM data is reported at a height of 10 metres, the observed 2 m winds were transformed to 10 m using the following logarithmic transform (Garratt, 1992),



$$u(z_2) = u(z_1) \frac{\ln \left[ (z_2 - d)/z_0 \right]}{\ln \left[ (z_1 - d)/z_0 \right]},$$

(1)

where $u(z_2)$ is the observed wind speed at height $z_2$, $u(z_1)$ the wind speed at height $z_1$, $z_0$ the roughness length to account for the effect of roughness of a surface on wind flow and $d$ is the zero-plane displacement to account for flow around obstacles. The bias correction was applied to the derived 10 m wind speeds, however, AWRA-L (Section 5) requires wind speed at 2m height, so Eq. (1) was again used to transform the near-surface wind speed back to 2 m height as input for the AWRA-L simulations.

## 4. The bias correction methods

Many bias correction (BC) techniques have been developed and several authors have provided detailed summaries and critiques of them (e.g. Fowler et al., 2007; Hewitson et al., 2014; Maraun, 2013, 2016; Maraun and Widmann, 2018; Teutschbein and Seibert, 2012), however, they fall into three general categories: (1) empirical scaling techniques, (2) non-parametric quantile matching (QM) and, (3) parametric QM (Potter et al., 2020).

Empirical scaling techniques include linear methods that preserve only the mean and apply a constant correction factor (Lenderink et al., 2007) while non-linear scaling, such as power transformations preserve the mean and variance (Leander and Buishand, 2007). Non-parametric QM techniques map the simulated quantiles of the cumulative distribution function (CDF) a variable to the observed CDF quantiles without any underlying assumptions that the variable can be modelled by a mathematical distribution, whereas for parametric QM, distributions are fitted to the variables before application of the QM (Heo et al., 2019). One of the advantages of using a parametric method is that it can provide a useful estimate based on the distribution when matching simulated fields that fall outside the calibration range of the historical observations (while noting that the extrapolation of a parametric distribution to values outside the range used to produce that distribution involves some uncertainties). For instance, the simulated field of projected temperature may not have any historical precedent, in which case assumptions have to be made as to how it will be associated with a quantile based on observations (Piani et al., 2010a, 2010b).

It is important to note that bias correction cannot add climate change information to that of the GCM to which it is applied. Rather, it modifies the coarse-scale GCM projections at a finer resolution to produce localised projections (Ekström et al., 2015). Furthermore, the application of bias correction may correct the bias at a local scale while ignoring the interdependency of variables, which has the unintended consequence of interfering with the underlying physics of the GCM (e.g. conservation of energy and mass) such that the resulting temporal and spatial representation of fields is unrealistic (Maraun et al., 2017).





Three statistical bias correction methods were applied to the GCM output variables. At a fundamental level, they are all variants of quantile matching (e.g. Maraun and Widmann, 2018), however, with important distinctions concerning their implementation. Despite the various pros and cons discussed in the literature of various BC methods, the choice was also (as for GCM selection) guided by pragmatic considerations, the foremost being ready availability to source code, thorough documentation in the scientific literature and access to technical support. Here we present a mostly qualitative overview of the individual methods; a complete mathematical formulation of the methods can be found in the respective citations.

### 4.1. ISIMIP2b

The Inter-sectoral Impact Model Intercomparison (ISIMIP) Project (https://www.isimip.org/) was designed to offer a consistent framework for projecting climate change impacts at different global warming levels across different sectors and spatial scales. Some of the sectors catered for in ISIMIP include regional water, fisheries, agriculture, biomes, and terrestrial biodiversity. To maintain consistency across so many diverse impact assessment models a common framework for bias correction and downscaling of GCM data was implemented. There have been several iterations of the bias correction method adopted for ISIMIP. At the commencement of the NHP the most recent version of the bias correction available for download was the one developed for the ISIMIP2b protocol. The complete description of the bias correction used in ISIMIP2b is a modification of the method described in Hempel et al., (2013) which extends the method described in Piani et al. (2010a, 2010b). Although ISIMIP is a set of protocols and output data across different impact models, we will refer to the bias correction as ISIMIP2b to be explicit. The ISIMIP2b method is classified as a parametric quantile matching (QM) method as it assumes variables of interest can be described by a particular parametric distribution. It uses Gaussian distributions to model daily temperature and gamma distributions to model daily precipitation probability distributions. For temperature, which can take both negative and positive values, the correction applied is additive, while in the case of fields such as precipitation, solar radiation, and surface wind speed, which have positivity constraints, the correction applied is multiplicative.

The ISIMIP2b method is comprised of the following steps: (1) correction of the monthly mean, (2) correction of the daily variability, and for precipitation (3) correction for the frequency of dry days and (4) correction for the intensity of wet days. Steps (1) and (2) have different implementations depending on whether the correction to be applied is additive (temperature) or multiplicative (precipitation, near-surface wind speed and solar radiation). Furthermore, steps (1) and (2) are designed to preserve the absolute trend of simulated temperature and the relative trend of the other variables. The preservation of the absolute long-term trend in the simulated temperature field preserves the climate sensitivity of the GCM and further ensures that there is consistency between the projected temperature and the bias-corrected climate change signal. For variables that have positivity constraints, the ISIMIP2b method conserves the relative trend. The use of an additive correction for temperature and a multiplicative correction for precipitation ensures that the ratio of the relative change in precipitation to an absolute change in temperature is conserved thus not altering the hydrological sensitivity of the GCM (Hempel et al., 2013).





### 4.2. MRNBC

The multivariate recursive nested bias correction method (MRNBC) is an extension of QM to include intervariable dependencies and in addition, corrects across multiple time scales. It was progressively developed from the nested bias correction (NBC; Johnson and Sharma, 2012) and recursive nested bias correction techniques (RNBC; Mehrotra and Sharma, 2012). The NBC corrects the distribution (mean and standard deviation) and persistence (lag 1 autocorrelation coefficient) at monthly, seasonal and annual time scales using a standard autoregressive lag 1 model (Srikanthan and Pegram, 2009). The RNBC method is an extension of the NBC method, whereby, the method is applied three to five times repeatedly so that it significantly reduces the biases in mean, variability and persistence related attributes in GCM/RCM simulations. The MRNBC method is a multivariate version of the above RNBC method. It simultaneously corrects many GCM/RCM variables, using multivariate first-order autoregressive model at daily, monthly, quarterly and annual time scales (i.e., like the NBC but multivariate) to impart observed distributional and persistence properties of the input fields (Mehrotra and Sharma 2015).

The implementation of a multivariate autoregressive model means that the MRNBC may better capture the joint dependence among input variables and hence be more effective in capturing the behaviour of natural processes that contribute to the variability of a field, particularly precipitation. Furthermore, as the MRNBC also corrects for the biases at seasonal and annual time scales, the bias-corrected atmospheric fields may provide a better representation of long-term variability in the downscaled simulations. This is of particular importance for hydrological applications where the representation of variability and persistence of precipitation is important for the simulation of occurrence and intensity of extreme events such as floods and droughts in downscaled simulations.

### 4.3. QME

The Quantile Matching for Extremes method is described in Dowdy (2019, 2020). As its name implies, it has a focus on matching the extremes of the CDF of a variable. It was originally developed to examine future projections of fire weather conditions for Australia based on the Forest Fire Danger Index (FFDI) (McArthur, 1967). The FFDI is a derived parameter, that uses daily maximum temperature, daily rainfall, afternoon relative humidity and afternoon wind speed in its formulation. Consequently, it uses similar variables as used in AWRA-L, with downwelling surface solar radiation also included here for this study (instead of relative humidity). In general, QM methods divide the quantiles of the observed and simulated fields into discrete bins (for example 100 percentile bins) so that events in, say, the top percentile of the simulated field are associated with those in the same quantile bin from the observational period. However, having uniform resolution quantiles does not capture extreme events well, especially those that fall outside the historical record. This method allows for simulated values that fall outside the historical reference values to be bias-corrected while still avoiding the influence of extreme outliers that may have occurred in the historical period. As an example, the temperature conditions recorded near Melbourne on 7 February 2009 when the Black Saturday bushfires occurred had no precedent in the prior 109 years of high-quality national temperature





records (Trewin et al. 2020). An exceptionally rare event that happens to occur in the training period for quantile matching may not be ideal to use in isolation for calibrating the model data.

In the QME method, an additional step is included to help represent extremes, where the top and bottom five most extreme
values are used to calculate the mean bias between the model and the reference data. By using the mean of the five extreme values (rather than fewer than that) the QME method is intended to provide a reasonable estimate of extremes up to about the 10–20 year ARI, while not being overly influenced by individual exceptional events that may sometimes occur in the training period. This is done individually for the top five values, as well as for the bottom five values, with those mean biases subtracted from those extreme values for the model data. The QME method is trained using the 43 years from 1975 to 2017, such that
estimates of extremes should be reasonably well-calibrated up to values of around the 10-year return period (i.e., average recurrence interval [ARI] near 10 years), based on a sample size of about four to five events on average for estimating those extreme values. Additionally, the mean bias for the five most extreme values is also subtracted from simulated values that lie outside the historical range of occurrence (noting the relevance of this in our changing climate).

**4.4. Effect of bias correction on GCM output**

Insight as to how the bias correction and the dynamical downscaling have modified the original GCM signal are shown in Figure 6 and Figure 7. Areal averages of the input and output data were produced for each ensemble member for all Australia and each NRM region. In addition, the original GCM data, spatially interpolated to the resolution of the AWAP, was archived to provide data before bias correction; we refer to this data as "NOBC" and "NOBC-CCAM" for the GCM and CCAM data,
respectively. Timeseries of maximum temperature for each GCM, averaged over all Australia and smoothed with a 10-year running mean are shown in Figure 6. The NOBC data (shown in magenta) is significantly cooler than the AWAP reference data (shown in black) and the bias correction has brought the GCM data in closer agreement to it. Consequently, the bias-corrected projections are warmer than the NOBC data for all GCMs. In contrast, the NOBC-CCAM data is warmer than the AWAP reference data and the application of ISIMIP2b has generally decreased the warming signal, however, CCAM-
ISIMIP2b is still significantly warmer than both the original GCM (NOBC) and bias-corrected data. The three bias correction methods are almost indistinguishable and only slight differences are discernible for the GFDL-ESM2M model.

A contrasting signal, shown in Figure 7, is present for precipitation. All the original GCMs, except ACCESS1-0, are significantly wetter than the AWAP reference data and that signal continues in the projections. The bias correction methods
have lowered the average daily Australia-average precipitation. The effect of the bias correction methods on precipitation shows more contrast among the methods than for maximum temperature. The NOBC-CCAM simulation is wetter than the reference data, however, the application of ISIMIP2b has produced a generally drier signal than when applied to the GCM data. The bias-corrected data correlates with both the NOBC and NOBC-CCAM data, however, it has translated the timeseries





to lower values. Finally, the application of a 10-year smoothing window demonstrates how the CCAM model produces quite distinct projections from its host GCM.

## 5.   The AWRA-L hydrological model

The Bureau's operational Australian Water Resources Assessment Landscape (AWRA-L) model was used to produce the Hydrological Projections outputs. AWRA-L is a daily semi-distributed water balance model (Frost et al., 2018) that is run daily using climate input data (pr, tasmax/tasmin, rsds, sfcWind) to produce hydrological outputs (runoff, evapotranspiration, soil moisture) at the 5 km × 5 km grid resolution. Land use and vegetation are represented within each grid cell by the fractional coverage of two hydrologic response units (HRUs) representing shallow-rooted vegetation (grass) and deep-rooted vegetation.

The terrestrial water storage component is represented by three soil layers, namely, top (0–10 cm), shallow (10–100 cm) and deep (100–600 cm) layers. Shallow-rooted vegetation is assumed to access soil moisture up to the shallow layer (up to 100cm) while deep-rooted vegetation is assumed to access soil moisture up to and including the deep layer (up to 600 cm). The soil moisture content modelled AWRA-L in the three soil layers is denoted by $S_0$, $S_s$ and $S_d$, respectively. For many hydrological applications the root-zone soil moisture ($S_m$), which is the sum of the soil moisture in the top two soil layers (0–100cm) is of

most interest (i.e., $S_m = S_0 + S_s$). Many fluxes and stores are output by the AWRA-L model, however, for the NHP, we focus on runoff ($Q_{tot}$), potential evapotranspiration ($E_0$) and root-zone soil moisture ($S_m$).

The AWRA-L model was calibrated by using 295 unimpaired catchments and validated against 291 evaluation catchments (Frost and Wright 2018). The median values of the relative bias, daily and monthly Nash-Sutcliffe Efficiency (Nash and

Sutcliffe, 1970) (NSE) were -0.01, 0.49, 0.67 respectively for runoff. Wasko et al. (2020) evaluated the performance of AWRA-L to simulate trends in catchment aggregated surface runoff and found good correspondence with observed trends in the southeast of Australia, in particular, the Murray Basin and Southern Slopes NRM regions. They also found reasonable correspondence between modelled and observed trends in streamflow in the northern parts of Australia for the warmer months. For the NHP, the AWRA-L model is forced using the historical daily gridded reference data set to output a gridded observation-

based data set of the 3 hydrological output variables ($Q_{tot}$, $E_0$ and $S_m$); this data set constitutes the reference hydrological data and, as for the GCM data, the historical period is defined as 1976–2005. To produce the historical AWRA-L hydrological data for the reference period (1976–2005), it was necessary to use historical GCM simulations beginning in 1960. This is because AWRA-L requires a "spin-up" time to equilibrate its water cycle since soil moisture is initialised at zero and is essential for modelling of deep-layer soil moisture which requires several years of simulation for surface water to drain into the deeper

levels. Starting the AWRA-L simulations in 1960 allows 16 years for the deep layer soil content to reach a pseudo-equilibrium before the beginning of the historical period corresponding to the gridded AWRA-L reference data.





## 6. Evaluation of the bias correction methods

Each of the bias correction methods has been published previously and comprehensively evaluated (see references in Sect. 4). A further detailed evaluation of both the GCM and AWRA-L output was performed of the historical simulations (1976–2005) against the gridded reference data sets (Vogel et al., 2022; submitted) They examined the spatial and temporally aggregated

(annual, seasonal and monthly) bias for each NRM region and multiple individual grid locations around Australia. A complete evaluation is outside the scope of this paper, however, we provide an overview of the spatial biases for the hydrologically relevant variables ($Pr$, $Q_{tot}$, $S_m$ and $E_0$) and annual and seasonal biases aggregated over Australia for all variables. The bias is calculated as the difference between the bias-corrected historical simulations and the reference gridded data. Figure 8 shows the annual and seasonal biases for water-related variables for ACCESS1-0 bias-corrected with the MRNBC method. Absolute

and relative biases are shown to aid the interpretation of the maps. For instance, an absolute bias of about 20 mm in the north of Australia during austral summer (DJF), which is also during the northern Australian "wet season", is of order −2.5% relative difference. Conversely, in southern Australia, absolute deficits of about 5–10 mm translate to relative deficits of 5%–25%. The mostly dry relative biases in precipitation manifest as corresponding dry biases in the runoff and root-zone soil moisture. Potential evapotranspiration most likely increases due to the increases in minimum and maximum temperature. Similar maps

(not shown) were produced for all 16 ensemble members.

The relative biases, averaged over Australia, are shown in Figure 9 (for water-related variables) and Figure 10 (for the remaining GCM output variables). Each subpanel corresponds to one of the bias correction methods implemented on each GCM. These summarise the information that was shown in the spatial bias maps (Figure 8). The bar plots presented here are

for Australia, however, similar analyses were conducted for each of the NRM regions to quantify their spatially aggregated bias and assess the contribution from each GCM and bias correction method. For instance, the application of MRNBC results in negative biases in rainfall, runoff, root-zone soil moisture and downwelling solar radiation for most GCMs on annual and seasonal timescales. The various model and bias correction combinations have differing bias magnitudes and signs between NRM regions. The reasons for this are due to the various GCMs' ability to simulate major climate drivers and also due to the

AWRA-L model to simulate the diverse hydroclimate of Australia. For example, each NRM will be dominated by differing land-use, agricultural and natural reserve regimes, which in turn impact the hydrological response in an NRM (Frost and Wright, 2018). However, it was found that the application of the bias correction methods, resulted in satisfactory confidence in the ability to simulate the hydroclimate across each NRM on annual, seasonal and monthly timescales.

## 7. National Climate and Hydrological Projections

The core data sets produced comprise gridded daily time series at 5 km × 5 km spatial resolution spanning the period 1960–2099. They are composed of bias-corrected climate data sets of precipitation ($Pr$), maximum and minimum temperature





(tasmax and tasmin), surface winds (sfcWind) and surface downwelling solar radiation (rsds) which were used to produce national hydrological projections of runoff ( $Q_{tot}$ ), actual and potential evapotranspiration *($E_{tot}$ and $E_0$)* and three layers of soil moisture ($S_0$, $S_s$ and $S_d$). For communication purposes, only projections of potential evapotranspiration are displayed, and soil moisture is referenced by root-zone soil moisture ($S_m$) which is the linear addition of the top two layers ( $S_m = S_0 + S_s$ ) and

represents the soil moisture in the top 1 m soil layer. For each variable, 16 ensemble members exist for each RCP, corresponding to the four GCMs with the three bias correction methods applied (12 ensemble members) and the CCAM RCM with ISIMIP2b applied (4 ensemble members). We will refer to the bias-corrected GCM data as input data and the AWRA-L simulations as the output data. The bias corrected climate input data complements the existing climate projections for Australia and moreover, can be used as input to other impact and assessment models.

To guide the interpretation of nationally consistent hydrological projections for Australia (and the NRM regions), several lines of evidence were explored; area-averaged transient time series, spatial change maps and area-averaged change plots for each RCP scenario and 30-year time slices. Here, we show examples of each of these, choosing the SSWF as an exemplar as its westernmost portion (southwest Western Australia) has a well-documented historical rainfall decrease signal (CSIRO and

Bureau of Meteorology, 2015). We also note that the rainfall declines in southwest Western Australia have a robust attribution to an anthropogenic influence (Timbal et al., 2006) and the scientific consensus is for those trends to continue (Dey et al., 2019). However, our intention is not to provide a complete hydrological overview of a particular region as these are provided elsewhere (see Sect. 8), but rather, to demonstrate the products and lines of enquiry that were developed.

### 7.1. Area-averaged time series

To examine the projections of the hydroclimate of a particular region, consider Figure 11, which shows projections for precipitation, runoff, soil moisture and potential evapotranspiration for the SSWF NRM region. Total area-averaged precipitation is expected to decline with concomitant declines in runoff and soil moisture, while potential evapotranspiration projections show increases. The ensemble statistics have been obtained by concatenating the historical and projected data and applying a 30-year running mean. This results in 15 years of data being excluded from the beginning and end of the timeseries,

such that they cover the period 1975–2085. This emphasises the long-term decadal-scale signal in the projections. Interannual variability, shown as yearly averaged data using the ACCESS1-0-ISIMIP2b ensemble member is shown to demonstrate the large departures from the long-term signal for all variables.

### 7.2. Spatio-temporal average change signal

Further information can be obtained by examining the change signal of future periods from the historical period (cf. Figure 4).

Spatial and temporal averages of the change signal were calculated for Australia and individual NRM cluster regions for each of the input and output variables. An example, for the four seasons and for SSWF is given in Figure 12, for precipitation, runoff, soil moisture and potential evapotranspiration where a general decrease in all variables is evident except for $E_0$. Some





general observations can be made: (1) there is a large spread among the ensemble members, however the spread among the individual BC methods is less than that of the individual GCM to which they were applied, (2) the change signal may be of opposite sign between the various GCM/bias correction combinations and (3) the median projected change is not necessarily monotonic, especially for the RCP45 scenario. We note that the median projected change signal for temperature (not shown) exhibits monotonic increases for all NRMS and both RCPs. In addition, we note that the (ensemble median) change signal for RCP85 for all variables is monotonic, however, that is not the case for all NRMs, and is likely affected by the strong change signal for precipitation in southwest WA (which is a subset of the SSWF NRM).

The first observation implies that in general, the choice of GCM contributes more to the ensemble spread than the application of the BC techniques and the inclusion of the CCAM model has added further diversity to the ensemble. The differing sign of individual members is a consequence of the model selection as indicated in Figure 3, whereby some GCMs differ in their wetting or drying signal. The non-monotonicity of the change signal (more apparent for other NRMs) could be due to differing multi-year time-scale processes (e.g. ENSO) represented by each GCM, especially in those areas where these processes dominate rainfall variability. Combined, these uncertainties make communication of the projections for end-users a challenging problem, however, we address some of these in Sect. 8.

The diversity of climate in Australia makes the use of annually-averaged changes unsuitable for most hydrological applications of the projections. As such, change plots were also constructed considering both four seasons (DJF, MAM, JJA, SON) and two seasons (a wet season; November–April and a dry season; May–October) to gain insight into the projected changes in seasonal hydrological projections. An example is given for SSWF in Figure 13. It can be seen that the major signal of the decrease in precipitation, runoff and soil moisture is due to their decreases in winter (JJA) and spring (SON), while the annual increase observed in potential evapotranspiration has no clear contribution from any particular season. The wetter signal is generally due to the MIROC5 and CNRM-CM5 models, however, the signal of CCAM forced with these models tends toward a drying of these signals. Note that in Figure 13, the annually averaged absolute (rather than relative) values are shown, indicating that small absolute changes may express as large percentage changes, particularly in low rainfall regimes. For example, the annual change in runoff ranges from approximately 20%–60% decrease, however, most of this is expressed as winter and spring runoff decreases of approximately 2–8 mm/year.

### 7.3. Time-averaged spatial plots

The final line of evidence used was that of time-averaged spatial plots, where the temporal average of the daily data for the historical and future time slices was calculated and the change (future minus historical) plotted. An example of the winter (JJA) ensemble median of variables is shown in Figure 14. They present the same temporal information displayed in Figure 13, however, include the spatial variability in the change signal. For instance, the decreases in precipitation, runoff and soil moisture for SSWF, displayed in the winter (JJA) sub-panels of Figure 13 are evident along the southern and western regions





of Australia. The spatial variability indicates that the change signal of decreases in precipitation, runoff and soil moisture is more prominent in the Western Australian portion of SSWF than in the South Australian sector. The projections for increased potential evapotranspiration apply to all time slices and RCPs. Similar plots were produced for all time slices and RCPs for annual and seasonal (both two and four) temporal aggregations for both relative and absolute changes.

### 7.4. Extreme wet events

The previous analyses have focussed on changes in the mean, whether it be the temporal and spatial mean (e.g. Figure 12 and Figure 13) or a temporal mean (Figure 14). Mean changes are useful quantities, however, for many impact studies, it is the quantification of events with a low probability and high impact (or extreme events) that are most crucial. As the climate warms,

heavy rainfall events are expected to increase due to the increased moisture content of the atmosphere (Sherwood et al., 2010). Model projections have also indicated that the frequency of extreme rainfall events in Australia is also expected to increase (Alexander and Arblaster, 2009; Perkins et al., 2014; Watterson et al., 2017). Extreme rainfall may be expected to manifest itself as an increased probability of flooding events, however, the risk of flooding is also determined by antecedent conditions soil moisture conditions. Wasko and Nathan (2019), suggested that in Australia (and other parts of the world), that following

heavy rainfall events soil moisture deficits are first augmented, such that, despite increases in heavy rainfall events there has been a decrease in flood magnitudes. Although it was also noted that soil moisture plays a decreasing role in flood severity as catchment size decreases (Wasko and Sharma, 2017). Here we examine some future flood scenarios based on changes in low probability, heavy rainfall and runoff events, which we denote as extreme events. Characterizing changes in flood frequency and intensity at a large spatial and temporal scale is challenging; flood risk is often dependent on local topography, sub-daily

rainfall intensity and antecedent conditions. A set of threshold-based indicators using precipitation and runoff have been calculated here to capture changes in flood risk on a broad scale. We have analysed these extreme rainfall/runoff events using the Generalised Extreme Value (GEV) distribution (Kharin et al., 2007; Perkins et al., 2014), where the changes are determined using the projected annual maximum daily rainfall/runoff, and the estimate of the 20-year return period of the annual maximum.

Figure 15 shows the results of the application of the GEV analysis in the SSWF NRM to the change in the daily mean, maximum and 1-in-20 return period rainfall/runoff event to the reference period (1976–2005). Even though the mean daily rainfall shows little change/slight decrease, the maximum daily and 20-year return period rainfall indicates a substantial increase under both RCPs and the increases are projected throughout the ensemble. The pattern of a decrease in mean rainfall

and an increase in rainfall extremes is found in almost all other NRM clusters and is supported by results from other studies (Alexander and Arblaster, 2009; CSIRO and Bureau of Meteorology, 2015; Wasko and Sharma, 2017). The projections for the extremes in runoff are less emphatic than those of rainfall, however, like rainfall, mean daily runoff shows decreases





(relative to the reference period) while maximum daily runoff and 20-year return period runoff indicate a substantial increase under both RCPs except for 2085 under RCP8.5. Similar analyses were conducted for each of the individual NRM regions.

## 8. Communication for end-users

To understand user requirements for the National Hydrological Projections service, a user-centred design process was undertaken using an expert consultant approach (Wilson et al., 2022). Over eight months, interviews with 56 potential users of NHP information, from 20 organisations (for which water availability was of paramount concern) across Australia were held. The organisations represented included: water utilities, government departments responsible for infrastructure planning, agriculture and water resources, hydro-electricity generators as well as emergency management services. They comprised existing customers of the Bureau of Meteorology and those anticipated being interested in using a hydrological projections service. Interviews were recorded and transcribed for documentation purposes. Following this process, key user requirements were grouped into three different types: 1) general information about the impacts of future climate change on Australian water resources in the context of historical hydro-climatic trends (general audience), 2) local and/or regional scale impact information including key messages of regional impact water resource assessment including guidance on the use of hydrological projections information (technical and policy audience) and, 3) access to application-ready hydrological projections datasets to be able to run their hydrological models and localised assessment analysis (planners and technical users).

To be able to attend to all three customer groups, we developed the following strategy: 1) the provision of application-ready National Hydrological Projections Datasets, 2) a user interface to provide a national picture of changes to key water variables and, 3) the provision of guidance material and regional hydroclimatic assessments.

### 8.1. Application-ready NHP dataset

The complete foundational NHP dataset is hosted on the Australian National Computational Infrastructure (NCI) via the NCI Data Collection and its Thredds server. The data is made publicly available free of charge under the following link: https://dx.doi.org/10.25914/6130680dc5a51 and under the Creative Commons — Attribution 4.0 International — CC BY 4.0. The dataset consists of 16 datasets per greenhouse gas emission scenario ordered by climate input variable and hydrological output variable (Table 2). Each variable is stored in individual netCDF files following a structured approach that is ordered by the host GCM, historical/RCP and bias correction method (including data with no bias correction applied). Metadata information is made compliant with CMIP5 metadata standards. The data is accessible either as OpenDAP, netCDF, HTTP, WMC or WMS data provision service.





### 8.2. Australian Water Outlook User Interface

A publicly facing portal was created by the Bureau of Meteorology called the Australian Water Outlook (AWO) service (https://awo.bom.gov.au) that connects people with interpreted key information on conditions and changes to water balance components. The Australian Water Outlook brings together information about the current and historical state of water in the

Australian landscape with near-term and seasonal forecasts, alongside information about the impacts of climate change on water resources (Wilson et al., 2022). The portal enables users to explore projections information in a meaningful way and selectable for their region of interest. Hydro-climatic changes are expressed for rainfall, evapotranspiration, soil moisture and runoff as absolute and relative change signals from the reference period of 1976–2005. The underlying NHP data is visualised as maps of mean changes for rainfall, runoff, soil moisture and evapotranspiration, for multiple future periods (30-year periods

centred at 2030, 2050, 2070 and 2085) and for the two greenhouse gas emission scenarios. The portal also provides the ensemble statistics showing the spread across the NHP member ensemble. The data in the interface is supported by regional reports (see Section 8.3) which contextualise the new information against the current climate, and the development of case studies and partnerships to demonstrate the potential application of the information. In addition, non-technical users are provided with digestible information on scientific and technical background information to the hydrological projections

through guidance material (e.g., FAQs and "About" sections).

Feedback from a range of key users during multiple stages of the portal development ensured that the information is fit for purpose. This co-design process also guided the user-friendly accessibility of the information presented. Maps, charts, information, and data can be freely downloaded from the AWO interface under the Creative Commons — Attribution 4.0 International — CC BY 4.0 license.

### 8.3. National hydrological projections assessment reports

To understand future impacts on Australia's water resources, we prepared eight tailored hydrological change assessment reports on plausible future changes in rainfall, potential evapotranspiration, soil moisture and runoff as well as including an analysis of wet and dry extremes conditions. These assessment reports are based on the eight clustered NRM regions (see Figure 5) and

use the information and analysis presented in Section 7. These NRM regions broadly represent groups of similar climate and biophysical settings in Australia and corresponding natural resources. The reports build on, and are consistent with, similar reports of CCiA, which provided comprehensive, robust, information on future climate changes for each of the NRM regions[4]. Our work builds a complementary picture in the context of changes to the regional hydrologic cycle and its future impacts.

The hydrological assessment region reports provide an overview of the key findings for each of the NRM regions. They describe each NRM region including background information about physiographic and hydro-climatic characteristics, recent

---

[4] https://www.climatechangeinaustralia.gov.au/en/overview/methodology/nrm-regions/




conditions and long-term hydroclimatic trends. The ability to simulate such changes is also examined by investigation of several questions: 1) are the climate models chosen able to represent the region's climate, 2) how well does the hydrologic AWRA-L model perform in the region, and, 3) how does the evaluation of the bias-correction methods affect the interpretation of the projections? Together, addressing these questions provides important context when assessing each NRM's hydroclimate.

Results from the 16-member ensemble (for each RCP) are presented as plausible future representations of change in the form of the magnitude in direction for each of the hydrological output variables and the two greenhouse gas emission scenarios, like the results presented in Section 7.

To demonstrate the applicability of the NHP data for future water resource impact analysis across Australia and to address the

uncertainty in hydrological projections, we use a storyline approach (Shepherd et al., 2018). The storyline approach describes an internally consistent evolution of plausible future events, allowing a way to focus on assessing only uncertainties that relate to a specific impact of interest. By establishing a set of storylines that represent plausible changes in risk, the impacts of a particular risk can be explored (as opposed to a forecast or likelihood of specific outcomes). They are effective for assessments where the change to risk is triggered by the interaction of multiple variables, such as the water supply and demand. For example,

applying the storyline concept to the Wet Tropics region in Northern Australia, we used the wet season runoff as an indicator for water storage (and hence supply) since almost all infilling occurs in the wet season. Changes to soil moisture in the dry season were used as an indicator of changes in demand due to increased agricultural irrigation, and domestic and consumptive use. Readers are referred to the Australian Water Outlook[5] to explore the regional assessment reports in further detail.

## 9.   Uncertainties in the hydrological projections

Climate and hydrological models are always an imperfect representation of reality (and plausible futures) and are therefore associated with various sources of uncertainties. There are several sources of uncertainty in the NHP, especially regarding the choice of GCM selection, emission scenarios and the bias correction and downscaling approach. For example, differences in the representation of natural variability (e.g., ENSO) in climate models can be a source of variation in the model's response to changes to atmospheric composition. Often, this is accounted for by examining many different models (an ensemble) to

understand the influence of the ways different processes are represented. There is also uncertainty associated with future human behaviours and policies around greenhouse gas emissions, which can be dealt with by examining multiple scenarios of emissions and human behaviour. Further sources of uncertainties stem from the influence of bias-correction approaches, which adjust model output towards desired characteristics of the observed climate; as well as from the hydrologic modelling and the representation of hydrologic processes itself.

---

[5] https://awo.bom.gov.au/about/overview/assessment-reports





A factor not considered in the NHP projections was vegetation responses to increased $CO_2$ and the resultant effects on projections of runoff. Offline models generally indicate an increase in terrestrial drying and drought metrics, especially those based on potential evapotranspiration (Naumann et al., 2018; Scheff and Frierson, 2015), and also predict decreases in runoff that exceed those projected by GCMs (Milly and Dunne, 2016). This has been attributed to potential evapotranspiration being

overestimated in offline models since they do not capture stomatal conductance reductions in response to elevated $CO_2$ (Roderick et al., 2015; Yang et al., 2019). AWRA-L was chosen as the hydrological model based on the evaluation and benchmarking of the available national models (Frost and Wright, 2018) and also for its ability to model projected hydrological variables when considering the larger uncertainties inherent to the climate models itself, as well as the static soil and vegetation inputs (Azarnivand et al., 2022). Importantly, those evaluations considered runoff, soil moisture and actual evapotranspiration

in the assessment of the models. However, AWRA-L is not a coupled model and was run independently using the bias corrected GCM climate data as input.

The lack of feedback between the GCM's and AWRA-L means that the potential role of increased CO2 levels on vegetation growth and evapotranspiration rates are not considered (Yang et al., 2019). It is also uncertain how important vegetation

feedbacks are in a mostly water-limited environment such as Australia, as the sensitivity of evapotranspiration parameterisation is more important in humid regions (Zheng et al., 2019a). In addition, recent observations indicate a robust increase in the partitioning of rainfall to evapotranspiration rather than runoff in response to the warming over land (Pascolini-Campbell et al., 2021). Finally, future land use change and vegetation change due to future temperature and water availability changes are also not considered and contribute to the uncertainties in the hydrological projections (Prestele et al., 2017). Despite these

caveats, we have developed these data sets and complementary analysis for the wider community, in particular, for those who wish to use hydrological projections for climate adaptation and those in the scientific community to further interrogate these data sets for research purposes.

## 10. Conclusion

This article has documented the National Hydrological Projections project (NHP) which resulted in the production of data sets

and communication of insights obtained from the data to end users. We developed a 16-member ensemble of bias-corrected GCM output which was used to force an offline hydrological model (AWRA-L). The ensemble was constructed by applying three bias correction techniques to the output of four GCMs and one of the bias correction methods to the output of the CCAM RCM forced by the same set of four GCMs. The four GCMs were CMIP5 model simulations (Taylor et al., 2012) using mid-range (RCP4.5) and high-emission (RCP8.5) scenarios (IPCC, 2000). The bias-corrected GCM variables were those required

as input to force the AWRA-L hydrological model, namely, precipitation, minimum and maximum temperature, surface wind speed and downwelling solar radiation. The subsequent hydrological output consisted of runoff, soil moisture and potential evapotranspiration. The bias correction methods were applied to historical (1960–2005) simulations using $0.05 \times 0.05$



(approximately 5 km × 5 km) gridded reference data at daily timescale. The resulting calibrations were then applied to the GCM output at a daily timescale for the period 2005–2099 and the subsequent AWRA-L simulations were produced on the same spatial and temporal scales. This is the first time that a nationally consistent set of hydrological projections have been produced for the entire Australian domain.

The bias-corrected GCM data and AWRA-L outputs were evaluated against historical reference datasets and shown to produce simulations that matched the historical record (i.e. the bias correction methods had allowed the GCM output to be suitable to force the AWRA-L hydrological model). Due to time and personnel constraints, only the output of four GCMs was able to be bias-corrected, however, they were chosen using guidance from the Climate Change in Australia report (CCiA; CSIRO and

Bureau of Meteorology, 2015), which suggested their inclusion for impact studies based on several criteria (see Box 9.2, p. 180 in CCiA). Furthermore, we explored the ensemble range of precipitation and surface temperature using the NHP subset of GCMs against the larger (40-member) ensemble of CMIP5 models used in CCiA and found they spanned a range of projections including wet/dry and cool/warm. This resulted in the NHP having ensemble medians and 10th–90th percentile ensemble spread of similar magnitude to the CCiA ensemble, although slightly cooler and wetter median projections of surface temperature and

precipitation. However, the application of the bias correction methods was found to apply corrections to the raw GCM data that were model and bias correction dependent and hence expanded the range of projections from consideration of the raw GCM data alone. Furthermore, the use of the CCAM RCM, which often had projections and temporal meanderings quite distinct from the host GCM, expanded the phase space of the original 4 host GCMs considerably.

To maintain some consistency with the CCiA report, we explored projections of precipitation, runoff, soil moisture and potential evapotranspiration for the Natural Resource Management (NRM) regions. For this paper, we used the Southern and South-Western Flatlands (SSWF) as an exemplar to present a range of analyses performed for the NHP. These included spatially-averaged time series, spatially and temporally averaged bar plots of the change signal (future minus historical) and temporally averaged plots of the spatial change signal. To construct the change signal, we considered four future time slices

of 30-years centred on 2030, 2050, 2070 and 2085 for RCP4.5 and RCP85 and analysed the magnitude and direction of change to provide detailed hydrological projections for each NRM. Readers are referred to the Australian Water Outlook[5] to explore the regional assessment reports in further detail. For further detailed regional analysis, guidance on the use of NHP data or further general information, please contact us via water@bom.gov.au.

## 11.  Code availability

The code required to implement each of the methods can be found in the GitLab repositories listed in Table 3. Note that these repositories are located on the NCI GitLab repository and are not available for general access. Please contact the listed author if you wish to gain access. QME code is somewhat different to MRNBC and ISIMIP in that it is currently only available for



the IDL software language and intended for experimental research purposes, with further details available by contacting this study's author (AD) directly.

## 12. Author contributions

RA, CD and AD conceived the original idea of the NHP. LW, CD, WS, PH, EV and JP devised the overarching research goals. JP, EV, WS, GK, V-CD, JR and LW managed data curation. JP, EV, WS, GK, SS, SBH, V-CD and JR conducted the formal analysis. RA and CD were responsible for funding acquisition. All authors contributed to the investigation process. JP, WS, V-CD, JR, FJ, RM, AS, VM, AO, MT contributed to the methodology. LW and UB-M oversaw project delivery. JP, EV, WS, GK, V-CD and JR wrote the software. EV, V-CD and JP conducted the validation. JP, EV, SS, WS, GK, VM, ZK, V-CD, AA and ST contributed to data visualisation. JP and SS wrote the original draft with contributions from VM, AO, MT, LW and U-BM. FJ, RM and AS supplied the MRNBC algorithm and contributed to validation. GB supplied the CCiA data.

## 13. Acknowledgements

This research/project was undertaken with the assistance of resources and services from the National Computational Infrastructure (NCI), which is supported by the Australian Government.

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





**Tables**

**Table 1: List of GCMs used in this study.**

| Model acronym | Institution | Horizontal resolution (lat × lon) | Reference |
|---|---|---|---|
| ACCESS1-0 | Commonwealth Scientific and Industrial Organisation (CSIRO) and Bureau of Meteorology, Australia | 1.25° × 1.875° | Dix et al. 2013 |
| CNRM-CM5 | Centre National de Recherches Météorologiques, France | 1.4° × 1.4° | Voldoire et al. 2013 |
| GFDL-ESM2M | GFDL, Princeton, New Jersey, United States | 2.0° × 2.5° | Dunne et al. 2012 |
| MIROC5 | Atmosphere and Ocean Research Institute and NIES, Japan | 1.4° × 1.4° | Watanabe et al. 2010 |

**Table 2: Overview of NHP foundational dataset variable available in the NCI Data Collection**

| Greenhouse gas emission scenario | Climate input variable | Hydrological output variable | Temporal and spatial resolution | Time period available |
|---|---|---|---|---|
| RCP4.5 / RCP8.5 | Rainfall, temperature maximum and minimum, wind, solar radiation | Runoff, soil moisture, potential and actual evapotranspiration | Daily grids at 0.05 × 0.05deg | 1960-2099 |

**Table 3: Description of code and repository information.**

| Code base | Description | Code location | Contact |
|---|---|---|---|
| MRNBC | Compilation of the MRNBC Fortran core. | https://git.nci.org.au/vd5822/mrnbc-processing | Justin Peter (justin.peter@bom.gov.au |
| | Data preparation scripts, job submission scripts and | https://git.nci.org.au/vd5822/mrnbc-processing | Justin Peter |



| | executables to run the MRNBC. | | |
|---|---|---|---|
| | The output from MRNBC is one text file per grid cell. Converts text files to netCDF and aggregates individual grid cells into a single file. | https://git.nci.org.au/jr6311/mrnbc | Justin Peter |
| | "Despeckling" routines to remove threshold values introduced by the MRNBC algorithm. | https://git.nci.org.au/vd5822/mrnbc-despeckle | Justin Peter |
| ISIMIP2b | Scripts and jobs needed to run ISIMIP2b. | https://git.nci.org.au/jr6311/isimip-bias-correction | Wendy Sharples (Wendy.Sharples@bom.gov.au |
| Transform winds | Transforms surface winds from 10m to 2m (and vice versa). | https://git.nci.org.au/vd5822/transform-wind-grids | Wendy Sharples |
| Compliance checker | Creates "compliant" copies of the netCDF files. Compliant means that the files conform to CMIP5 standards[6]. | https://git.nci.org.au/vd5822/compliant-invertlat | Wendy Sharples |

---

[6] See Sect. 4 of https://pcmdi.llnl.gov/mips/cmip5/requirements.html for details of CMIP5 compliant data requirements.





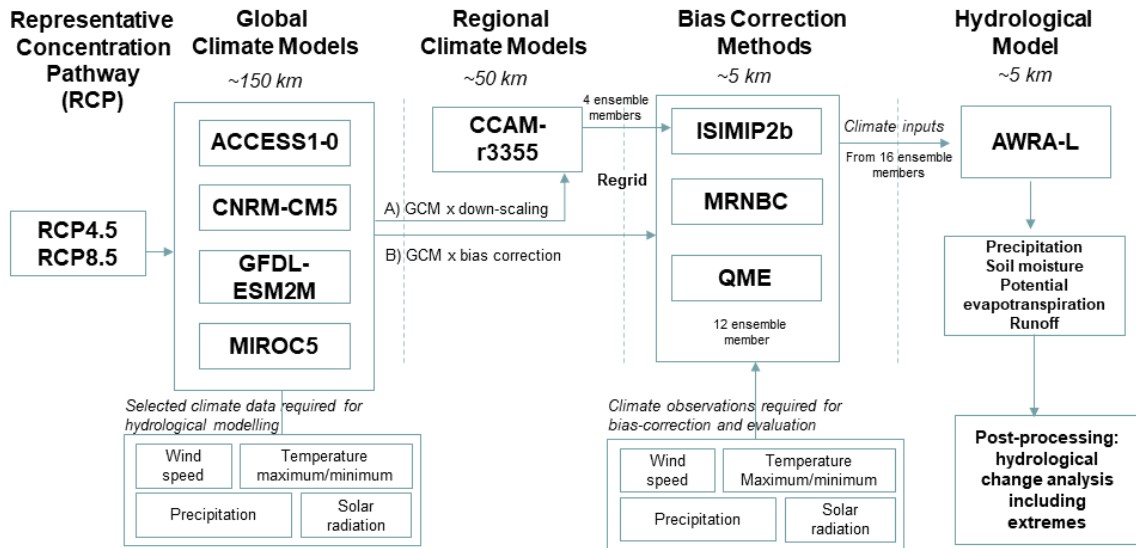

**Figure 1: Modelling flow diagram for the National Hydrological Projections. Individual components are detailed in the main text.**

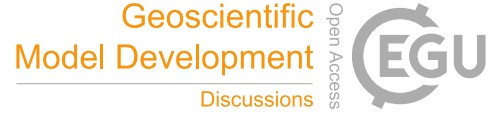

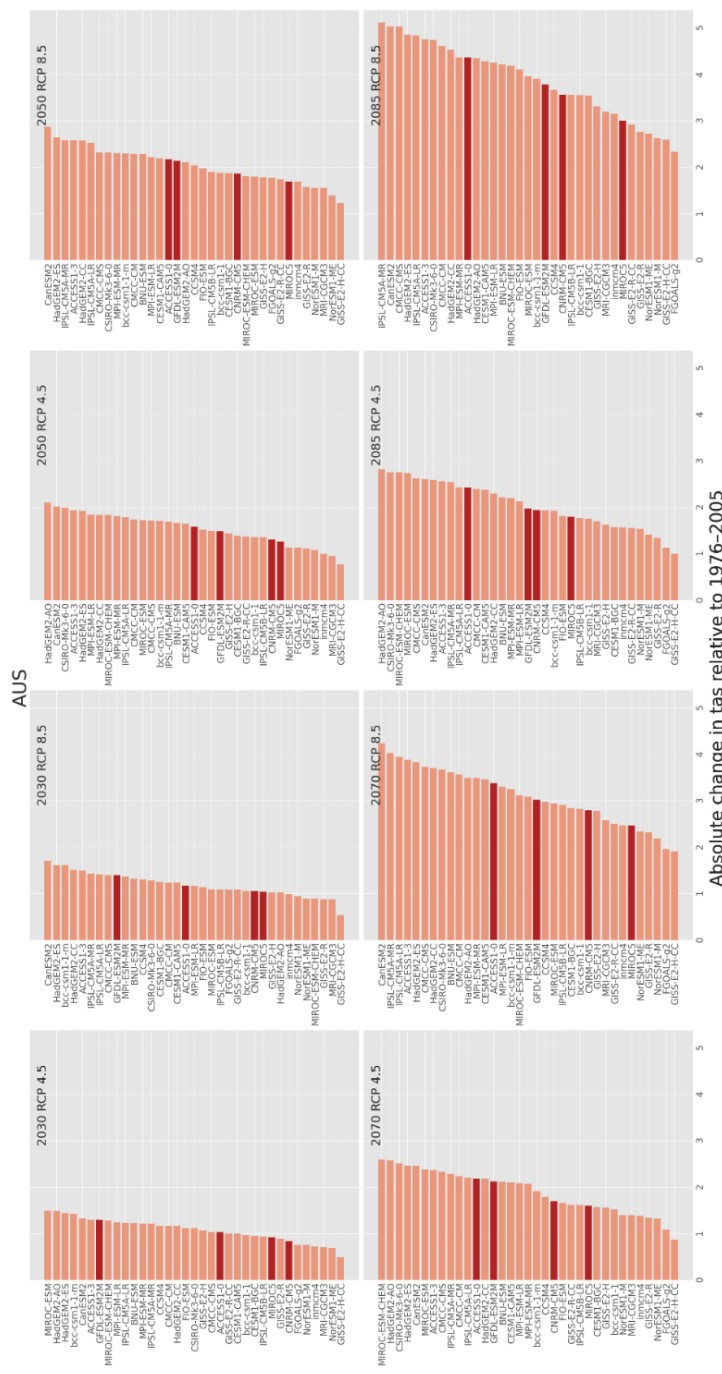

**Figure 2: The ranking of surface temperature (tas) for the GCMs used in this study with regard to the CCiA ensemble. The horizontal bars indicate the change signal (the difference of the nationally-averaged quantity from the climatology for the period 1976-2005). Four 30-year periods are shown centred on 2030, 2050, 2070 and 2085, and for RCP4.5 and RCP8.5.**



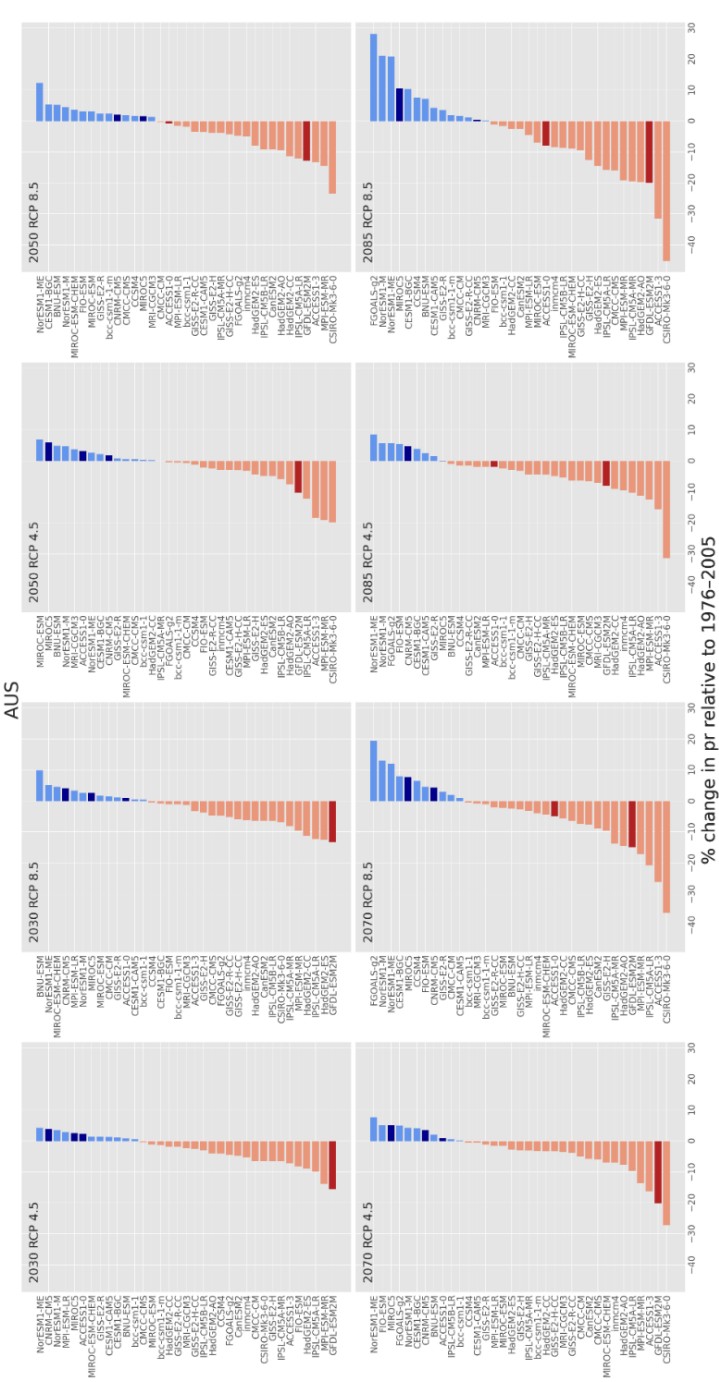

**Figure 3: Same as Figure 2 but for precipitation. Note that percentage change in precipitation is shown.**



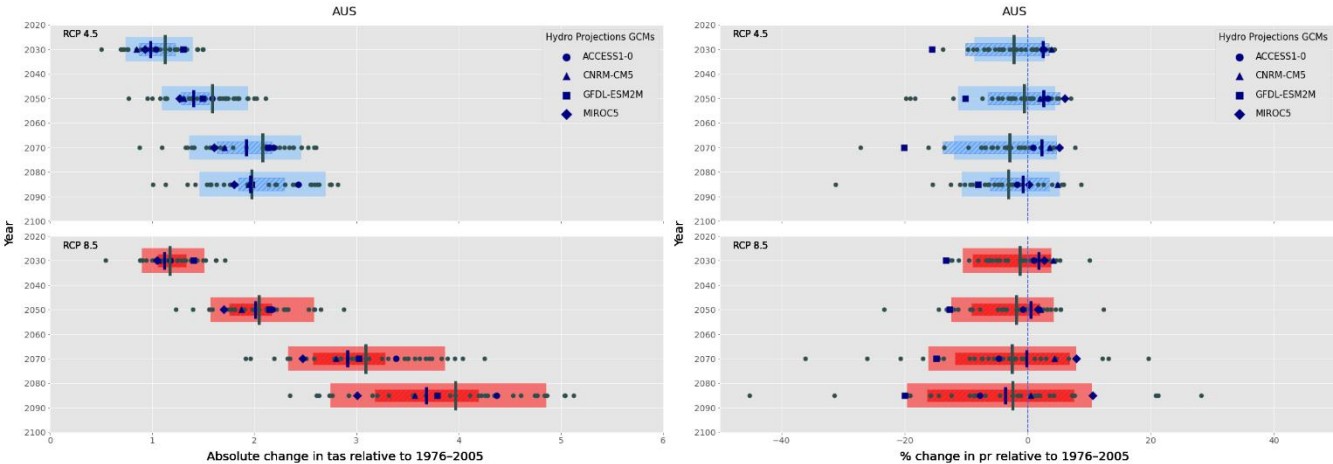

**Figure 4: Change plot of surface temperature (left) and precipitation (right) for all of Australia. The NHP models are indicated by the larger blue symbols while the CCiA models are shown as smaller grey-filled circles. The thick vertical lines represent the ensemble median and the left and right extent of the rectangles the 10th and 90th percentile of the model ensemble, respectively. Time is displayed on the ordinate and the magnitude of change (from the historical period) on the abscissa; RCP4.5 is shown in blue and RCP8.5 is shown in red. The larger rectangles and grey vertical line and dots show the spread of the CCiA ensemble while the smaller hatched rectangles, blue vertical lines and shapes display the NHP ensemble.**






**Figure 5: The eight NRM regions used in the NHP to present results relevant to the differing climate regimes across Australia. From: (CSIRO and Bureau of Meteorology, 2015).**





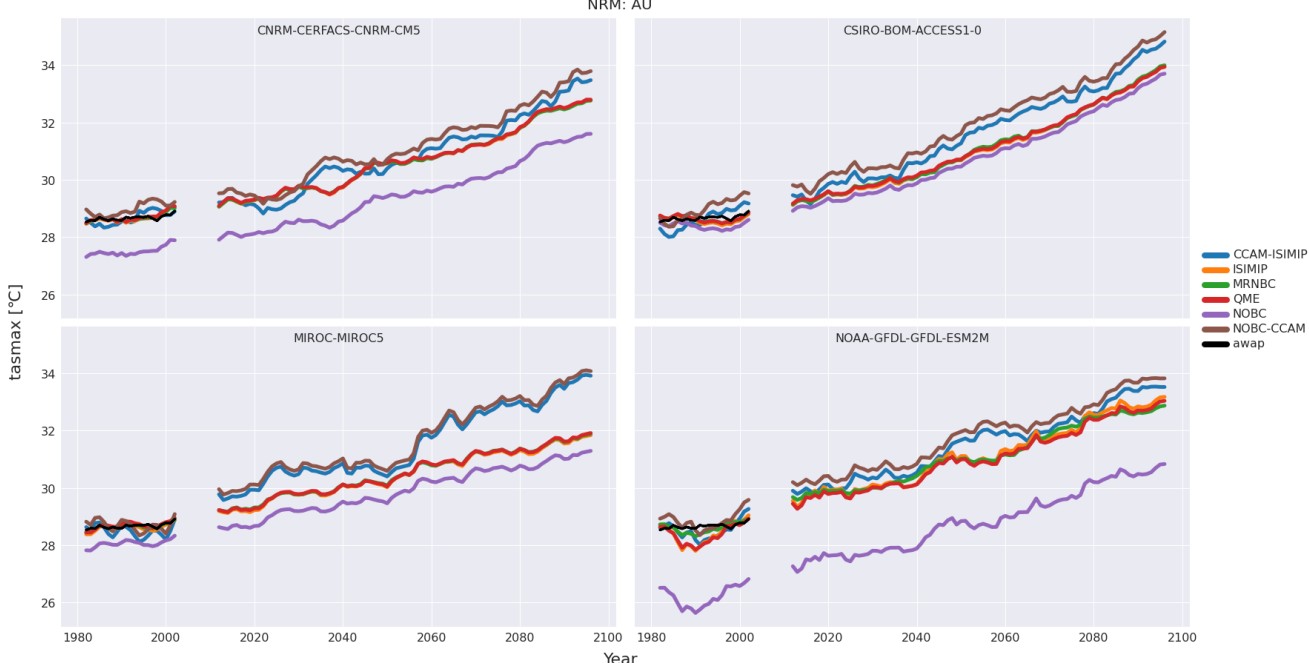

**Figure 6: Timeseries of maximum temperature averaged over Australia for each GCM in the ensemble. Different colours correspond to the bias correction methods. The labels NOBC and NOBC-CCAM indicate the GCM and CCAM data before the application of a bias correction. Timeseries have been smoothed with a 10-year running mean.**



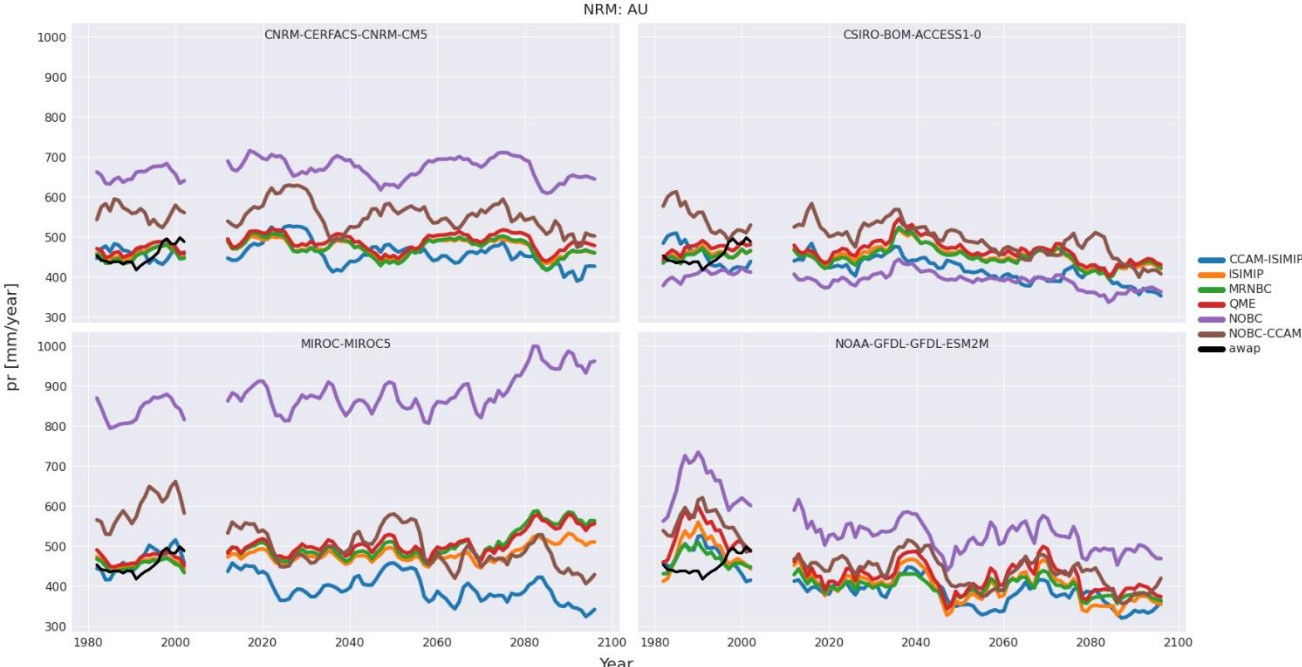

Figure 7: Same as Figure 6 but for precipitation.



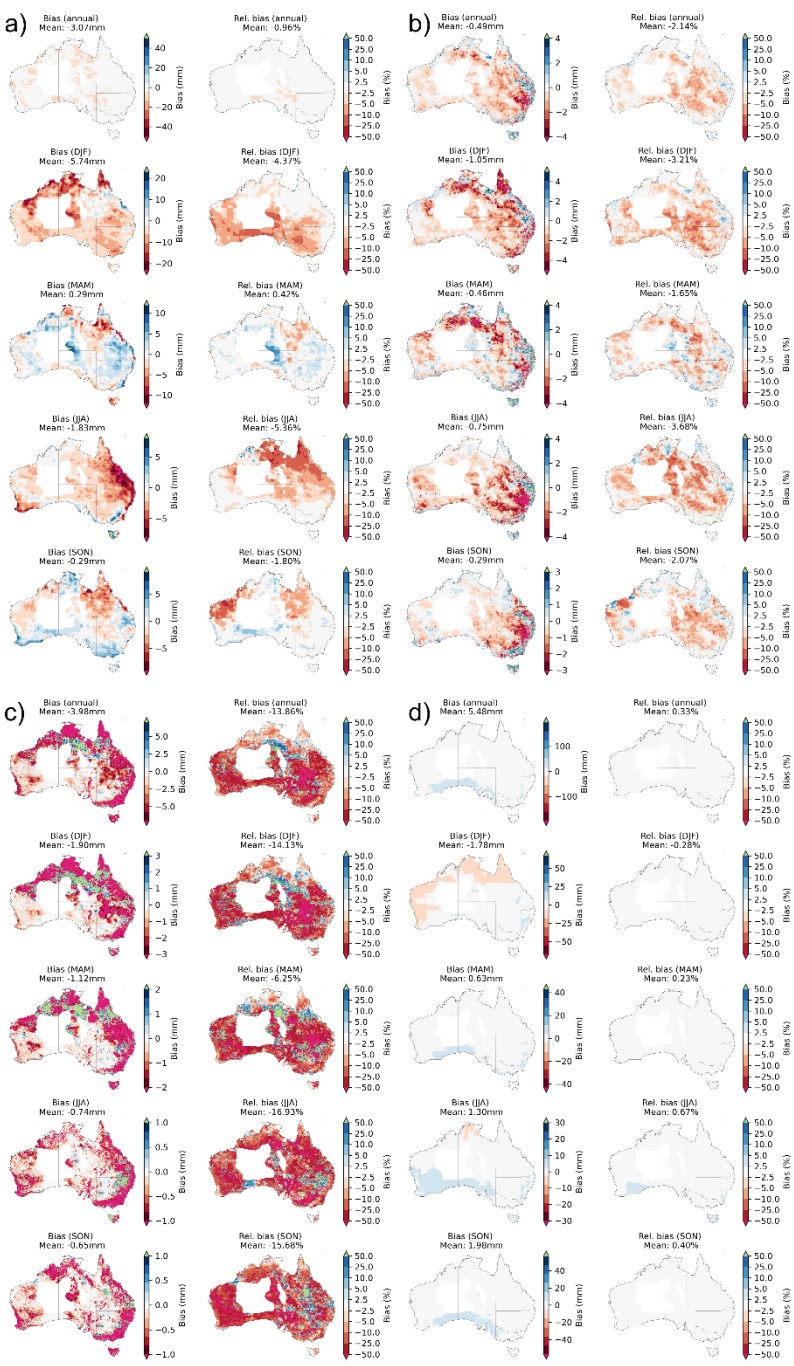

**Figure 8: Bias maps of annual and seasonal precipitation (a), runoff (b), root-zone soil moisture (c) and potential evapotranspiration (d). The maps are for the ACCESS1-0 GCM with the MRNBC bias correction method applied. The bias is the difference between the historical (1976–2005) GCM simulations from the historical (1976–2005) gridded reference data. The mean bias is shown at the top of each map in the panel. Note that the scale for the absolute biases is dynamic, while for the relative bias it is fixed in the range -50%–50%.**



**Figure 9: Bias histograms for all of Australia for precipitation (a), runoff (b), root-zone soil moisture (c) and potential evapotranspiration (d). Annual and seasonal (four seasons) biases are shown. The legend indicates the various bias correction and GCM combinations.**





**Figure 10: As for Figure 9 but displaying the biases for maximum temperature (a), minimum temperature (b), downwelling solar radiation (c) and near-surface wind speed (d).**





Figure 11: Projected ensemble change in precipitation, runoff, root-zone soil moisture and potential evapotranspiration for RCP4.5 (blue) and RCP 8.5 (red) scenarios in the SSWF NRM region. The shaded area represents the 10th to 90th percentile range for all ensemble members. The time-series for ACCESS1-0 (rcp8.5) is included after 2020 to show the variability projected for an individual ensemble member. The reference gridded observations are shown prior to 2020. The ensemble has been smoothed with a 30-year running mean.

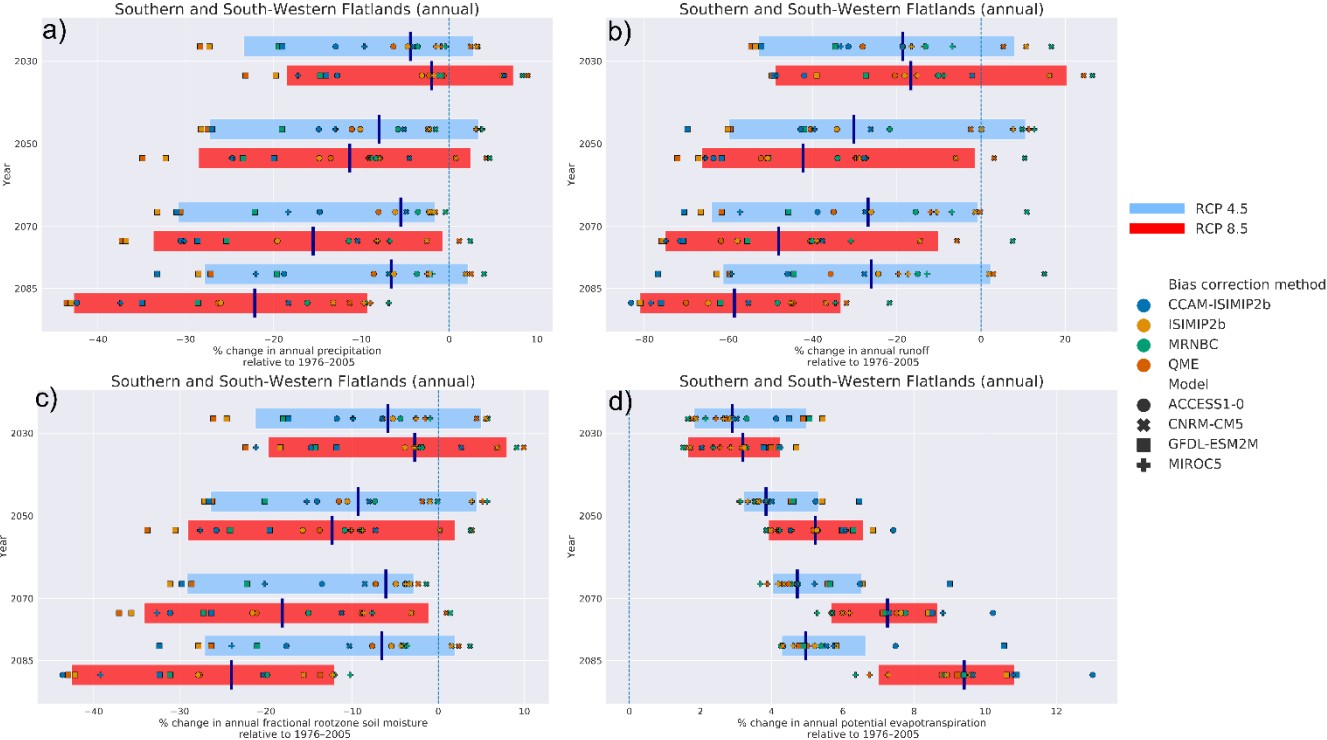

**Figure 12: Relative change of annual (a) precipitation, (b) runoff, (c) root-zone soil moisture and (d) potential evapotranspiration. Changes are indicated for each GCM and BC method combination for four 30-year time periods centred on 2030, 2050, 2070 and 2085. The change is the difference of the future from the historical period (1976–2005) expressed as a percentage. The left and right extent of the shaded boxes represent the 10th and 90th percentile of the GCM/BC ensemble and the vertical solid line shows the median (50th percentile). RCP4.5 is shown in blue, RCP 8.5 in red. The ensemble spread for each RCP has been offset from the centre from of our 30-year time slices for legibility, however, each box represents the average of the variability centred on the time slice indicated on the ordinate. Note that time is descending on the ordinate and the change (in percent) is shown on the abscissa.**



**Figure 13: As for Figure 12, but displaying seasonally-averaged absolute change values for four seasons (DJF, MAM, JJA, SON) for each variable; precipitation (a), runoff (b), root-zone soil moisture (c) and potential evapotranspiration (d).**





**Figure 14: Ensemble median relative change in winter (JJA) (a) precipitation, (b) runoff, (C) root-zone soil moisture and (d) potential evapotranspiration. The change signal is only shown for the 2030 and 2070 time slices for RCP85.**





**Figure 15: Changes (%) in mean daily, maximum daily and 20-year return period daily rainfall (a) and runoff (b).**