# Peer review of "Continental-scale bias-corrected climate and hydrological projections for Australia"

_Geoscientific Model Development, 2023_

## Author Comment (AC3)

GMD-2023-07 Anonymous Reviewer 1:

The work to produce these projections represents a mammoth effort with a large interdisciplinary team. This project/data set will have a lasting legacy with a large number of potential applications, and I hope the potential benefits of using this data set are realised by the community so as to
5  reward the authors on their hard work. The projections have been well described and evaluated extensively. The choice of GCMs and downscaling methods has been done with care and are appropriate to the outcomes of the project. Finally, the projections themselves are of great interest to the hydrologic (among others) community. Please see my comments below which at first may seem extensive but are relatively minor and can often be treated as suggestions rather than being
10  prescriptive.

Thank you for this comment. It is very much appreciated.

General comments:

15  # Some of the figure legends/axes were a bit small and hard to read.

Many of the figures have been updated and/or redrafted. In particular, some figures have been removed from the main figure and placed in a supplementary figures section. This applies to Figures 2, 3, 8, 11, 12 and 13. In addition, high resolution EPS format figures have been supplied for publication.

20

**The abstract describes succinctly the product development and how it was performed. I wonder if a sentence at the start of the abstract on the need for the product might help with context.**

We have modified the second sentence to: "The NHP aimed to provide nationally consistent hydrological projections across jurisdictional boundaries to support planning of water-dependent
25  industries. NHP is complementary to those previously produced by federal and state governments, universities, and other organisations, for limited geographical domains."

**I know it was mentioned, but it just wasn't quite clear to me how the 9am-9am data from Australia was matched to the GCM data (which I assume is 12am-12am).**

30  The GCM data are simulations that are not based on observations for any given day in particular, such that the timing of an individual event in the simulations (such as a rain-bearing storm, an El Nino event, etc.) is not intended to relate to the timing of a similar individual event in observations. However, the climatology (e.g., average statistics based on long-term data aggregation) can be matched between the GCM and observations data, as is done in this study, as this doesn't rely on an
35  individual day in the model data being matched to an individual day in the observations. This means that for this study's purpose, it doesn't matter what time of day the daily rainfall observations are based on, as long as the model and observations data both use daily aggregation, which is the case.

We tried to explain how we reconciled the timing of the observations with that of the projections in Section 3.1. Since we are dealing with *projections* (as opposed to predictions), the temporal
40  synchronicity (at least at daily timescale) between the model and of observations is of little

relevance. However, the following sentence (from our paper), hopefully explains it best: "Note that, while observations are made from 9 am to 9 am, the bias corrections are calculated by calendar day."

5  # Page 5, Line 15: Clarification for me please - Is it usual to only use SSTs as the forcing from the GCM in CCAM? I understand CCAM doesn't have lateral boundary conditions making it quite unique – is my understanding correct?

You are correct, the CCAM model has a stretched grid with a focus over a region of interest, so is more an actual GCM, rather than an RCM, which uses as its forcing the output of the GCM for its
10  lateral boundary conditions.

CCAM can be run in two major configurations. The first uses spectral nudging of the winds to force CCAM back to those generated by the host GCM and in that configuration tends to follow the projections from the host GCM more closely. In the configuration used for NHP, the forcing from the GCM comes from bias correcting the SST's and CCAM is allowed to develop its own climate and the
15  subsequent projections can be quite different from the host GCM.

We have modified the text (P.5, L8-12):

" CCAM can be forced via two methods, one using bias-corrected sea surface temperatures (SST's) and another using spectral nudging to update the circulation to that of the host GCM. It was the
20  former configuration used for NHP, whereby the mean and variance SSTs of the host GCM are bias corrected to provide boundary conditions for CCAM to produce 50 km resolution projections of the atmospheric state over the Australian continent (Clarke et al., 2019; Hoffmann et al., 2016)."

**Section 2.2: I don't think the authors should change their text, but as a comment, it felt the GCM**
25  selection was given 1-2 lines of attention on Page 4 and then two pages of attention was given to how the GCM projections fit within the ensemble of GCMS. This felt a little unbalanced to me. I understand it is important to show the spread of possible futures and how this ensemble covers it, but some text sounds like the authors justifying that 'only' four GCMs are sufficient. In particular on Page 7, Lines 9-15 almost seem to defensive to me, and I don't see a reason why the authors need to
30  defend 'only' four GCMs when they do in fact represent a good range plausible future. Moreover, I think the authors analysis is superior for the fact that they considered the best GCMs for Australia (rather than using all GCMs blindly).

Thank you for this comment. We tried to emphasise that GCMs spanned a broad range of the phase space of the GCMs recommended by CCiA and moreover, the complete range of CMIP5 GCMs.

35

**Page 5, Line 21: It feels odd to state the method not used was spectral nudging when the method that was used wasn't stated?**

Please see the answer to the previous comment (Page 5, line15) above.

Page 7, Line 9: "uncertainties are underestimated" – which uncertainties? Should this be "uncertainties in the GCM choice are underestimated"?

We have modified the sentence to:

" Nevertheless, the spread of the NHP ensemble is less than that of CCiA, suggesting that the NHP GCM ensemble spread may be less than the CCiA GCM recommendations."

**Page 7, Line 25: "calibrate the GCM output" I would prefer the word calibrate to not be used, also calibrating data doesn't quite make sense. Can this be reworded please?**

"Calibration" changed to "bias correct".

**Page 8, Line 20. Up to the authors if they want to keep this sentence, but AWAP is gridded and will by definition underestimate point data. It is true also if one looks at catchment averages for extremes AWAP is slightly biased down but (to me anyway) the differences aren't great. See Figure 5 in Nathan, R., Jordan, P., Scorah, M., Lang, S., Kuczera, G., Schaefer, M., Weinmann, E., 2016. Estimating the exceedance probability of extreme rainfalls up to the probable maximum precipitation. J. Hydrol. 543, 706–720. https://doi.org/10.1016/j.jhydrol.2016.10.044**

We wanted to point out some of the biases of the AWAP data set. We modified the sentence: " Furthermore, in areas of steep topography…" to " Furthermore, in areas of steep topography and a sparse gauge network,…", to emphasise that the AWAP gridded analysis is subject to increasing disparity between point observations in data-sparse regions and topographically complex regions.

**Page 9, Line 25: The description sounds more like downscaling "modify coast-scale GCM projections at a finer scale" rather than bias correction. Maybe some rewording in this paragraph would be appropriate?**

Anonymous Reviewer 3 made the point that using bias correction techniques for down scaling should be a key point. However, we are hesitant to use the term "downscaling". Please refer to our response to their point (page 3, line 24). The bias correction does provide provide fine-scale information compared to the GCM/RCM input, so we think this is appropriate terminology.

**Page 12, Line 24: When you say decreases the warming signal it sounds like it has decreased the trend, but to me to the trend before and after bias correction of the CCAM data (brown and blue lines) is identical? So maybe some rewording here is necessary. See the comment below.**

Sentence modified to: "…the application of ISIMIP2b has generally decreased the warming bias…"

**Page 12, Line 27: Maybe I am taking exception with the word "signal". That implies to me some sort of temporal trend, but here you just talking about the GCM being wetter, which isn't a signal, it's just bias. Apologies about the long comment – maybe just changing the word from signal to bias would be beneficial?**

Agreed, we have changed "signal" to "bias" (in this section)

**Figure 8: Because you summarise seasonal results in Figure 9 and Figure 10, Figure 8 could just have the annual results only to make the figure more manageable? I know Vogel et al., 2022 has an extensive evaluation of the bias correction, but I think one figure just for one variable (say runoff) with all the bias correction methods would really be beneficial (can just be for one GCM) – given the amount of time spent outlining the bias correction methods (and their potential impact on the results).**

We have removed the seasonal plots from this figure and placed them in the supplementary figures section.

**Page 14, Line 32 (and elsewhere): Sometimes precipitation is pr, and sometimes it is Pr (in italics).**

All instances of Pr changed to "pr". The same has been done for qtot, e0, etot, so, ss and sd, where sometimes we had used capitals with subscripts.

**Figure 11: Bottom panel missing units on the y-axis?**

The units have been included for the bottom panel.

**Page 16, Line 19: Did you use the wet and dry season? I think Figures 13 and 14 just use the regular DJF etc seasons?**

We didn't show the wet-dry season as this NRM (SSWF) is in a temperate climate zone and subject to the four seasons. We used the wet/dry season analysis to construct analyses for the Assessment Reports (https://awo.bom.gov.au/about/overview/assessment-reports) described in Section 8.3.

**Page 16, Line 9: Not sure, but I know of work that found that the bias correction method was the greater contributor to the ensemble spread. Not sure if the authors have comments on why the different results? See Wasko, C., Guo, D., Ho, M., Nathan, R., Vogel, E., 2023. Diverging projections for flood and rainfall frequency curves. J. Hydrol. 620, 129403. https://doi.org/10.1016/j.jhydrol.2023.129403**

Was that because that evaluation was for extremes? We have shown for the mean state.

**Figure 15: Given that AWRA-L is a water balance model, has it been evaluated for extremes and if not can a comment be made on its applicability for this purpose. The above manuscript and the following found a possible underestimation of extremes or the change signal in changes for extreme events. Ho, M., Nathan, R., Wasko, C., Vogel, E., Sharma, A., 2022. Projecting changes in flood event runoff coefficients under climate change. J. Hydrol. 615, 128689. https://doi.org/10.1016/j.jhydrol.2022.128689**

Yes, an evaluation of the ability of the AWRA-L model's ability to simulate extremes and climate variability has been undertaken, coming to the conclusion that AWRA-L is reliable and accurate enough to be able to simulate the wide range of plausible projected outcomes (Azarnivand et al., 2022).

**Section 7.3: I wonder if the "maps" came first (Section 7.3 was Section 7.1), then it would make an easier transition to Section 7.1 and Section 7.2. Looking at the maps, you see the strongest change in SSWF and then you can drill down on the results for that region. My other concern with just focussing on the JJA season. Most rainfall occurs in the summer in the tropics so the results presented here aren't as meaningful as they could be – I guess I would prefer these maps to be annual – and to be the first item displayed in Section 7. This would also follow better as again Section 7.4 focuses on SSWF.**

We followed the structure that was presented in the Assessment Reports, in particular Section 4 of the reports (e.g. https://awo.bom.gov.au/about/overview/assessment-reports#regionsandreports). We appreciate the suggestion of the reviewer, however, we wanted to demonstrate the lines of reasoning that were followed in constructing the assessment reports for the Australian Water Outlook.

**Section 8: Am I right in saying that temperature projections are not available as part of the Australian Water Outlook Service but are available on NCI? I feel temperature is an important variable for example when calculating fire risk, and one that many other users would be interested in.**

Yes, that is correct. The AWO service was designed with water availability its central focus. However, we of course hope that the bias corrected climate variables will be used by others for differing impact studies. Temperature and other climate variables are available as application-ready datasets as per NCI Data Collection.

**Section 8.3: Line 27 confused me a bit – would it be better to have a link to the reports here (instead of the end of Section 8.3)?**

We tried to emphasise how the assessment reports build on the foundational work performed in CCiA. We think that the confusion is due to the footnote in line 27 being to the CCiA reports. First we describe the construction of the assessment reports and then provide a link as a footnote. As a compromise, we have made the link to the assessment reports explicit in the text rather than providing them as a footnote.

**Section 9: The first paragraph could almost be removed, and the section relabelled "Limitations".**

Thank you for this comment. We have deleted the first paragraph (which was a rehash of the uncertainties of the NHP data set) and relabelled this section "Limitations of the hydrological projections".

Page 22, Line 7: I wonder if "due to time and personnel constraints" could be rephrased with "due to the large spatial domain…" it is clear (to me anyway) that you couldn't be expected to use more GCMs than you already have due to the large domain and sheer scale of the project.

Yes, although it was a factor (time and personnel constraints) we like the recommendation of the reviewer. Furthermore, another reviewer (number 2) mentioned a similar issue.

Editorial:

Page 2, Line 11: "…south-east with changes in streamflow typically…" might read better.

Changed as recommended.

Page 3, Line 5: missing a space after the reference.

5 Modified.

Page 5, Line 24: Doesn't have to be bold and can be part of the paragraph.

Thanks for spotting. Text formatted as "normal" rather than "heading 2".

Page 7, Line 9: extra new line.

Modified.

10 Page 19, Line 22: Change from 3$^{rd}$ person to 1$^{st}$ person with "we". Could revert to be consistent with the rest of manuscript.

Modified to use "we" as recommended.

Page 20, Line 23: I think there is a track changes mark under the apostrophe in "model's".

This paragraph (and the rogue track changes mark) was deleted as per the reviewer's previous
15 recommendation. (See #Section 9 comment).

Page 21, Line 12: CO2 (subscript the 2)

Modified.

**References:**

20 Azarnivand, A., Sharples, W., Bende-michl, U., Shokri, A. and Srikanthan, S.: Analysing the uncertainty of modelling hydrologic states of AWRA-L – understanding impacts from parameter uncertainty for the National Hydrological Projections., 2022.

Clarke, J., Grose, M., Thatcher, M., Hernaman, V., Heady, C., Round, V., Rafter, T., Trenham, C. and Wilson, L.: Victorian Climate Projections 2019 Technical Report., 2019.

25 Hoffmann, P., Katzfey, J. J., McGregor, J. L. and Thatcher, M.: Bias and variance correction of sea surface temperatures used for dynamical downscaling, J. Geophys. Res. Atmos., 121(21), 12,877-12,890, doi:10.1002/2016JD025383, 2016.

---

## Author Comment (AC4)

The authors document the development of a national set of hydrological projections for Australia. They provide good motivation for the development of the product/service and provide extensive description and evaluation of the output. My main comment is around the use of the 3 bias correction methods to expand the ensemble. It's not clear to me why they looked at 3 methods and what it adds. E.g. at one point (for temperature) the authors describe how "the three bias corrections methods are almost indistinguishable…" (pg 12, line 9, Figure 6). Also at pg 16, line 9 (re Figure 12), the authors comment: "The first observation implies that in general, the choice of GCM contributes more to the ensemble spread than the application of the BC techniques". I think some more discussion and justification for the inclusion of all 3 bias correction methods is required.

It wasn't apparent how the various bias correction techniques would affect the projections when they were first chosen. We decided on using three techniques due to their availability: QME was developed at the Bureau, we had established connections with UNSW to obtain the MRNBC code and the ISIMIP2b method is freely available and was specifically designed to be used for impact studies. A thorough evaluation and "ranking" of the BC methods is available (Vogel et al., 2023). This study found that some methods may be more appropriate depending on the application. For instance, the QME may perform better at capturing the extremes, while the MRNBC was found to perform better overall, especially for the hydrological output.

      Reviewer three asked a similar question, so for completeness we will include our response to them
here:

          • " Page 16 Line 9-10: Do the results imply that perhaps only one best performing bias
            correction technique is needed for your application? Perhaps more GCMs and/or RCMs
            should be included to better gauge the uncertainty of the future projections.

      That may indeed be the case. For instance we found that QME performed best when measured
against extremes, while the MRNBC performed the "best" overall when ranked across a range of metrics (Vogel et al., 2023). It would be ideal to include a full suite of CMIP models, or at least, a subset based on benchmarking criteria. That opportunity was not available to us for NHP (as explained in Section 2). In any case, the frequentist approach may not be the best way in which to represent uncertainty, particularly for impact assessment studies (Shepherd, 2021).

There are emerging opportunities however for the Australian Climate Service (ACS) to produce similar data sets based on a carefully selected subset of CMIP6 models (Grose et al., 2023). One of their key findings is that: "The projections cannot be considered a probabilistic or balanced estimate of uncertainty given the limited ensemble size and underlying epistemic uncertainties. The ensemble can however be used in a 'climate futures' or 'storyline' approach to illustrate
plausible future climates that broadly span the range of possibilities suggested by CMIP6, while producing added value at the regional scale." The assessment reports produced for the NHP (https://awo.bom.gov.au/about/overview/assessment-reports) were also structured with a storyline approach, which we document in Section 8.3."

      We have modified the text (now page 12; lines 29-32) to:

" The three bias correction methods follow very similar trajectories and only slight differences are discernible for the GFDL-ESM2M model. Since temperature is a smooth field (both spatially and temporally) and all the bias correction methods are variants of quantile matching, this indicates that at least for the mean and large geographical aggregation, the bias correction imparts less uncertainty than GCM selection in the projections."

Other points:

Page 5, line 1-2: I respect that pragmatic choices need to be made but I think the authors should make some comment around the validity of RCP8.5 for future risk assessment e.g. https://www.nature.com/articles/d41586-020-00177-3

We tried to address this with the following sentence:

"These emissions pathways were chosen to provide a high (RCP8.5) and moderate (RCP4.5) set of temperature projections, noting that the set of modelled greenhouse gas emission pathways provided in CMIP5 have relatively minimal deviation before 2050 and the observed climate change trends for $CO_2$ emissions and temperature in recent decades indicate that the high emissions pathway (RCP 8.5) has been followed more closely than other emissions pathways (e.g., RCP 2.6)
(Schwalm et al., 2020; Stocker et al., 2013)."

We have added: "However, we also note that RCP 8.5 may be at the high end of emission scenarios for future risk assessment (Peters and Hausfather, 2020).

Page 5, line 24: Why is this sentence in bold?

This has been fixed.

Figure 2: Perhaps it's obvious, but I think somewhere in the caption the authors should write that the darker shaded bars indicate selected models

The caption has been modified to: "*The ranking of surface temperature (tas) for the GCMs used in this study (darker shading) with regard to the CCiA ensemble (lighter shading). The horizontal bars*
*indicate the change signal (the difference of the nationally-averaged quantity from the climatology for the period 1976-2005). Four 30-year periods are shown centred on 2030, 2050, 2070 and 2085, for RCP8.5."*

Page 6, line 1: The use of 1976-2005 historical period is a departure from CCiA. Can the authors
comment why they chose that period?

We chose that end year (2005) based on the year the CMIP5 historical data finished (like CCIA). Additionally, we chose a reference time period of 30 years since our end goal was to produce hydrological projections, where we wanted to try and capture some long period hydrological features. For instance, a 20-yr reference period may not be of sufficient length to capture drought
periods.

Page 7, line 10: Why not include the CCAM simulations to get a better picture of the spread relative to CCiA?

This analysis was to show where our selection of GCMs sit within the CMIP5 ensemble. Since CCAM is a stretched grid model forced by a GCM (see (Thatcher and McGregor, 2009) it carries much information from the forcing GCM. It is true that in the configuration used for NHP (using bias-corrected sea surface temperature), that the interior of CCAM is allowed to evolve freely (i.e., it can develop a climate quite distinct from the forcing GCM) it still contains information from that GCM. The intention was to show the spread of the selected GCMs within the CMIP5 ensemble, particularly given that the selected GCMS were a subset of those recommended in CCIA. The spread resulting from the inclusion of the CCAM is explained in further analyses, particularly that of Section 7.

Page 7, line 21: GCM/RCM not just GCM

GCM changed to "GCM and/or RCM simulations…"

Page 7, line 23: This sentence doesn't make sense. I think the "are" before precipitation should be replaced with a colon or dash.

Thank you for spotting this. The "are" before precipitation has been replaced with a colon.

Page 7, line 26: Does this need to be updated to AGCD?

At the time of producing preparing the observational data, the data was known as AWAP. It is now known as AGCD (v1.0) / AWAP. We have added a footnote: " The interpolation used to produce the AWAP analysis is currently being updated by the Bureau of Meteorology and is now known as [Australian Gridded Climate Data (AGCD) / AWAP; v 1.0.0](). See (Evans et al., 2020) for details."

Page 9, line 15: This sentence doesn't make sense and needs to be rewritten.

Sentence now reads: "Non-parametric QM techniques map the simulated quantiles of the cumulative distribution function (CDF) to the observed CDF quantiles without any underlying assumptions that the variable can be modelled by a mathematical distribution, whereas for parametric QM, distributions are fitted to the variables before application of the QM".

Page 12, line 26: As I understand it, the purpose of an ensemble member is to add new information. How are the 3 bias correction methods adding new information? Can the authors comment on this? (see my main comment)

Figure 6, 7: Perhaps I missed it but why are CCAM-MRNBC and CCA-QME not included?

This was due to time constraints. The ISIMIP2b method was the first to be implemented during the NHP and was already available when the CCAM output was also available, while the other two methods were still under development. We have added a footnote (footnote number 5) in Section 4 (Page 10, line 1):

" Due to time constraints, the only bias correction algorithm applied to the CCAM output was the ISIMIP2b method."

Fig 8: Although it's mentioned in the text, I think it would be helpful to add a sentence in the caption about why there are data gaps in the maps.

The following has been added to the caption: "Data sparse regions have been masked (see Section 3.1)".

Page 16, line 5: NRMs (or NRM regions) not NRMS

NRMS changed to NRMs.

Figure 12: It seems pointless to label each of the 4 plots with "Southern and South Western Flatlands" – this could just be written in the caption. I think having the variable (e.g. precipitation, soil moisture etc.) clearly visible at the top of each plot would be helpful.

We have removed the NRM description (Southern and South Western Flatlands), however, we have not included the variable in the title but rather in the figure caption to be consistent with the other plots.

Figure 13: It's very difficult to interpret these plots, can the resolution be sharpened?

We appreciate that there is a lot of information in these plot. We considered only showing one
variable and including the other variables in the supplementary material; however, it is important to show the seasonal characteristics for all the hydrological variables in the main manuscript. Our other option would be to have each panel as a separate plot, however, that would increase the figure numbers (and we had reviewer comments about trying to reduce the number of figures).

We have included a high resolution eps file to examine the finer details of the plots (e.g. examining
specific bias correction methods and/or GCM). The explanation in the text is mainly concerned with the ensemble mean interpretation, which is illustrated in the low-resolution (png format) version.

Page 17, line 13/14: You've written "antecedent conditions soil moisture conditions". I assume you mean "antecedent soil moisture conditions".

Yes, modified accordingly.

Page 22, line 8: It seems strange that 'personnel issues' is listed here but not earlier in the manuscript.

Reviewer 1 had a comment about this phrase and it has now been modified (following the recommendations of reviewer 1) to:

" Due to the large spatial domain, only the output of four GCMs was able to be bias-corrected…"

**References:**

Evans, A., Jones, D., Smalley, R. and Lellyett, S.: An enhanced gridded rainfall dataset scheme for Australia. [online] Available from: http://www.bom.gov.au/research/research-reports.shtml, 2020.

Peters, G. P. and Hausfather, Z.: Emissions - the "business as usual" story is misleading, Nature,
577(January), 618–620, 2020.

Schwalm, C. R., Glendon, S. and Duffy, P. B.: RCP8.5 tracks cumulative CO2 emissions, Proc. Natl. Acad. Sci. U. S. A., 117(33), doi:10.1073/PNAS.2007117117, 2020.

Stocker, T. F., Qin, D., Plattner, G. K., Tignor, M. M. B., Allen, S. K., Boschung, J., Nauels, A., Xia, Y., Bex, V. and Midgley, P. M.: Climate change 2013 the physical science basis: Working Group I
contribution to the fifth assessment report of the intergovernmental panel on climate change., 2013.

Thatcher, M. and McGregor, J. L.: Using a scale-selective filter for dynamical downscaling with the conformal cubic atmospheric model, Mon. Weather Rev., 137(6), 1742–1752, doi:10.1175/2008MWR2599.1, 2009.

Vogel, E., Johnson, F., Marshall, L., Bende-Michl, U., Wilson, L., Peter, J. R., Wasko, C., Srikanthan, S.,

Sharples, W., Dowdy, A., Hope, P., Khan, Z., Mehrotra, R., Sharma, A., Matic, V., Oke, A., Turner, M., Thomas, S., Donnelly, C. and Duong, V. C.: An evaluation framework for downscaling and bias correction in climate change impact studies, J. Hydrol., 622, 129693, doi:10.1016/J.JHYDROL.2023.129693, 2023.

---

## Author Comment (AC5)

This article introduces the development of a national hydrological projections (NHP) service for Australia, including the choice of GCMs and RCM, application and evaluation of three bias correction methods, and driving the Bureau's landscape water balance hydrological model (AWRA-L) to produce hydrological projections. This national hydrological service provides valuable information on the impact of climate change on hydrological cycles over Australia to end users. The overall structure of the manuscript is coherent while wordsmithing is necessary, especially in the first half of the article. Besides, I have a few comments and suggestions for authors to consider.

Thank you for the recognition of the NHP service.

**Specific comments**

Page 2 Line 22-26: It is mentioned here that Australian states may prefer to use their own downscaled projection products. Key issues are that data are too heterogenous for use across intersect jurisdictional boundaries, and clear instructions are not provided. These issues are addressed in the NHP service, but are users in these states now tend to use your products rather than use state operated ones? Could you give some insights into this point?

Thank you for this question, it is an important one. We don't consider the NHP projections to be a replacement for those produced by states (or other jurisdictions) but rather a supplement. As we explained in the paper, the current projections are too heterogeneous for use across boundaries and the current projections often stop at those boundaries.

There has been uptake of the NHP data sets, in particular the Energy Sector Climate Initiative (ESCI),
which was initiated to provide information on energy security by the Australian Energy Market Operator (AEMO), CSIRO and The Australian Bureau of Meteorology. Two case studies were investigated by ESCI: projections of runoff for hydroelectricity production (Hydro Generation (climatechangeinaustralia.gov.au)) and projections of soil moisture for infrastructure (Soil moisture & infrastructure (climatechangeinaustralia.gov.au)). These are examples where state projections
would have not been able to supply the necessary data, since the National Energy Market (NEM) operates in Queensland, New South Wales, Victoria, Australian Capital Territory, South Australia and Tasmania (see AEMO | National Electricity Market (NEM)).

The NHP data was also used by the Western Australian government, who wanted to investigate projections of runoff in the Pilbara region. The Western Australian government does not have local
hydrological projections and so utilised the NHP data sets. In consultation with water managers, conditions for "drought" and "wet" years were defined and the NHP data used to identify changes at the catchment scale. An example is shown in the figure below:

[Figure]

*Figure 1: The NHP based assessment of change in drought frequency vs change in wet year frequency for each model (markers), bias correction method (colours), and emissions pathway (marker fill) by 2050 for the Yule catchment.*

Using this information, it was shown that changes in drought frequency and changes in wet year frequency are highly correlated at the catchment scale.

The above are two examples of how the NHP data has filled the gap that would not be enabled by the current state-based projections. However, we hope that users who have access to state-based hydrological projections will use the NHP data sets to expand their projections.

Page 3 Line 1: I could not fully understand what transient projection means. Please give a clearer definition and/or example. Also, repeated word 'applied' in footnote 1.

Previous projections (using scaling methods) have only been provided for a discrete time corresponding to the length of the observational baseline period (for example 30 years), rather than continuous projections up to 2100. We have replaced "transient" with "continuous".

Repeated word "applied" has been deleted.

Page 3 Line 24: According to the context, you simply interpolate GCM and RCM to 5km spatial resolution before applying the bias corrections. I reckon using bias correction techniques for downscaling should be the key point here.

We are hesitant to use the word "downscaling" when applying bias correction. Although the term is used, we consider downscaling to either be statistical or dynamical. In the former, statistical relationships between the large-scale GCM output (e.g. 500 hPa winds) and the historical observations are derived and subsequently applied to the GCM projections to downscale the GCM projections to a local scale. Dynamical downscaling is the use of an RCM to produce the finer scale projections. It is true that the bias correction has provided finer-scale resolution and picked up important features, in particular, cooler temperatures in elevated topography in the Great Dividing range and various coastal rainfall features. However, the output of RCMs will still have biases (as demonstrated by the CCAM output for NHP). The first part of this point (referring to downscaling) was in reference to the use of CCAM, not the use of bias correction a downscaling technique.

Another point I am interested in is whether you have tried only applying simple mean (additive or multiplicative) correction to the GCM outputs to drive hydrological model, and using sophisticated methods to correct hydrological outputs. What is your rationale of bias correcting climate outputs prior to driving the hydrological model? Even though the multivariate bias correction accounts for the inter-variable, temporal and spatial structure of the model outputs, the bias adjustment process may have changed temporal features of the model series.

It is well documented that a large proportion of the biases along the projection impact modelling chain come from the GCM data itself (Azarnivand et al., 2022; Bosshard et al., 2013; Dobler et al., 2012; Giuntoli et al., 2015; Joseph et al., 2018). Therefore, it makes sense to bias correct GCM data rather than impact model outputs. Secondly, a full (both spatially and temporally) land surface dataset is needed to bias correct the hydrological outputs which is not available for Australia. Thirdly, the available options for hydrological reanalysis across Australia to bias correct the outputs do not match in the AWRA-L model historical dataset both in accuracy and reliability (Frost et al., 2018). Even so, we have evaluated the AWRA-L model's ability to simulate extremes and climate variability, coming to the conclusion that AWRA-L is reliable and accurate enough to be able to simulate the wide range of plausible projected outcomes (Azarnivand et al., 2022).

With regard to changing the temporal characteristics (for instance, wet-dry spell length), we are currently examining the NHP data to investigate how the bias correction has modified these. We hope that others examine these features, especially those related to "extremes" (by which we mean the upper percentiles) to determine how the bias correction has modified these.

Page 4 Line 8: This is the first time AWRA-L model is mentioned in the introduction. I think more descriptions of AWRA-L are needed in this section because the choice/development of hydrological model is definitely an important part of the NHP project.

The introduction served to outline the motivations for initiating and developing the NHP project. In the dot points prior, we outline the major decisions made, in a sequence that we developed and applied during the various stages of the NHP project. We did consider the use of other hydrological models (e.g. GR4J), however, the AWRA-L was settled on, primarily because the expertise was available at the Bureau to run it and it also the Bureau's operational model. We think that outlining the major decisions (as in the dot points) and then describing them in detail is appropriate.

Page 4 Line 21-26: In line 24, what does 'variation between CMIP5 models' mean here? The temporal variance and climatological mean? Regarding the GCM selection, I would like the authors to explain more about how you narrow down the selection from 8 to 4 GCMs. You mentioned that all required variable data are available among 47 CMIP5 models, and CCiA recommended 8 models. What are your criteria to choose these four CMIP5 models out of eight. In addition, I am curious why you include an RCM to increase the ensemble range, and four RCM simulations are only corrected using one bias correction technique. Why not simply include other four GCMs recommended by CCiA?

"Variation between CMIP5 models" means those that were considered to range between hot/cool and wet/dry projections. We have modified the sentence to read:

 "… as well as to provide a reasonable representation of the wet/dry and cool/warm variation between CMIP5 models…".

Figures 2 and 3 are too small. Please consider redo them into a 4 rows × 4 cols plot. Figure 8 is also too small. Please consider split it into two or more plots.

We have modified Figures 2 and 3, to only show RCP8.5 and moved the RCP4.5 figures to Supplementary Figures.

Page 6 Line 26: What is your rationale of calling these four GCMs a 'reasonable' subsample of the CCiA ensemble? Please specify.

We have removed the word "reasonable".

Page 7 Line 25: The bias correction methods, ISIMIP2b and MRNBC, are trained over 1976-2005. Is it because the wind speed observations start from 1975? The QME method is trained over 1975-2017, 10 which is 13 years longer. Please clarify and comment on to what extent the use of different training period would affect the bias corrected climate variables, and further the hydrological projections.

One of the main aims for producing hydrological projections in this study was to use a 30-yr time period to help capture more detail of the long-term variability than a 20-yr time period would. The starting year of 1976 was chosen as some studies have shown there was a climate shift that occurred 15 in Australia (particularly in the south-west of the continent) in the mid-70s (e.g. Hope et al., 2010).

The time-period of 1975-2017 for calibration of the QME was used to capture some recent extreme meteorological events (e.g., the Black Saturday bushfires in 2009 and the Queensland floods in 2010/2011) including with this method originally being designed to have a key focus on details for extreme cases. A thorough evaluation of these contrasting bias correction methods and a ranking 20 against several criteria has recently been published by Vogel et al. (2022). For instance, the QME performed better than the other two methods when evaluated against 5-year maxima and extreme percentiles (but less good in some cases for other metrics such as mean runoff), which may relate to several aspects of the QME algorithm and time periods used. For example, the QME method was applied for 3-month seasons as part of its aims around maximising the sample size for extremes, as a 25 complementary approach to the application for individual months for the other methods used in this study). Further details about the choice of the 1976-2005 reference time period can be found in the Australian Water Outlook FAQ page (https://awo.bom.gov.au/faqs/projections).

Page 8 Line 25: Before 1990, daily climatological averages (for each day of the year) are used for 30 solar radiation. How did this affect the training of the bias correction models as the 'true' values are not recorded?

The use of the daily climatological values was a pragmatic decision made due to the non-availability of measurements before this time. It is difficult to know exactly how the use of the climatological values has affected the overall bias correction methods. However, we note that the evaluation of 35 Vogel et al. (2022 see Table 3), indicated that the MRNBC (a multivariate method) performed particularly well when evaluated against hydrological output metrics. This indicates that a multivariate technique that considers the joint marginal dependencies of the distributions (including solar radiation) has performed better than the univariate methods, further indicative that the use of climatological solar radiation values has not been detrimental to the projected hydrological outputs.

Page 12 Line 18-19: This statement could be moved to before Section 4.1, where the data required for the bias correction is introduced.

We have moved this as suggested (now near the beginning of page 10):

"Three statistical bias correction methods were applied to the GCM output and one (ISIMIP2b)[1] to the CCAM output (see **Error! Reference source not found.**). In addition, the original GCM data, spatially interpolated to the resolution of the AWAP, was archived to provide data before the application of bias correction; we refer to this data as "NOBC" and "NOBC-CCAM" for the GCM and
CCAM data, respectively."

Page 13 Line 27: Do you train bias correction model using the period from 1976, and apply the trained model to correct climate model simulations from 1960 to 2099? Please clarify this in the text.

Yes, that is what we did. The sentence has been modified:

"To produce the historical AWRA-L hydrological data for the reference period (1976–2005), it was
necessary to use historical GCM simulations (bias corrected to the reference period) beginning in 1960."

In Figure 8c, despite small absolute biases, the relative biases for root zone soil moisture over all four seasons are very large compared to other variables. Will the large relative biases of the soil moisture lead to inaccurate information for the community?

Please see our response to the comment below. This was a mistake in the drafting of the figure on our part, where we mixed panels (b) and (c) of Figure 8.

Page 14 Line 17: Figures 9 and 10 are plotted without any interpretations. Please give some comments/explanations on these results. From my understanding, part of Figure 9 shows area-averaged relative biases presented in Figure 8. However, the results of MRNBC-ACCESS1-0 in Figure
9c contradicts those in Figure 8c, where the averaged relative biases should be at least <-10%. Moreover, in these figures, these relative biases or bias values may not be representative because the negative and positive values may be cancelled out in the area averages. I suggest plotting averaged absolute biases without signs across Australia and NRM regions.

This was a mistake in the drafting of the figure. Thank you for spotting this error in the review. The
numbers are now consistent, however, there are slight differences. This is due to the different ways the biases were calculated in the spatial plots (Figure 8) compared to the bar plots (Figures 9 and 10). In the former, area-averaged biases were calculated by first calculating the relative bias (%) for each grid cell and then averaging it over Australia. For the bar plots, the Australia-wide mean was calculated first and then the relative bias (%) calculated from the two regionally averaged values.

The manuscript has been modified to reflect this (page 14, lines 20-32 and page 15, lines 1-3).

Page 14: It would be better to show the bar charts for NRM regions (similar to Figure 9 and 10), at least SSWF, in the manuscript or in the supplementary material.

Figures S7 and S8, which are for the SSWF NRM have been included in the supplementary material.

Page 15 Line 23-24: Why not showing the ensemble statistics using the yearly averaged data instead
of 30-year running mean? I suspect the range of 10th and 90th percentiles over time will not be too messy.

We show the 30-yr running mean to extract any trends in the data and, in particular, to show differences between the RCP8.5 and RCP4.5 scenarios. We do include and example of one model in the relevant Figure (Figure 11) to illustrate one possible future and the role of year-to-year
* * *
[1] Due to time constraints, the only bias correction algorithm applied to the CCAM output was the ISIMIP2b method.

variability. The analysis produced was designed to have correspondence with that of CCiA (CSIRO and Bureau of Meteorology, 2015; see Figure B.6.2.4, Pg. 84). The model ensemble can be a good representation of the projected multi-decadal trends, however, is of limited value to explain yearly variability.

Page 16 Line 9-10: Do the results imply that perhaps only one best performing bias correction technique is needed for your application? Perhaps more GCMs and/or RCMs should be included to better gauge the uncertainty of the future projections.

That may indeed be the case. For instance we found that QME performed best when measured against extremes, while the MRNBC performed the "best" overall when ranked across a range of
metrics (Vogel et al., 2023). It would be ideal to include a full suite of CMIP models, or at least, a subset based on benchmarking criteria. That opportunity was not available to us for NHP (as explained in Section 2). In any case, the frequentist approach may not be the best way in which to represent uncertainty, particularly for impact assessment studies (Shepherd, 2021).

There are emerging opportunities however for the Australian Climate Service (ACS) to produce
similar data sets based on a carefully selected subset of CMIP6 models (Grose et al., 2023). One of their key findings is that: "The projections cannot be considered a probabilistic or balanced estimate of uncertainty given the limited ensemble size and underlying epistemic uncertainties. The ensemble can however be used in a 'climate futures' or 'storyline' approach to illustrate plausible future climates that broadly span the range of possibilities suggested by CMIP6, while producing added
value at the regional scale." The assessment reports produced for the NHP (https://awo.bom.gov.au/about/overview/assessment-reports) were also structured with a storyline approach, which we document in Section 8.3.

Figure 14: Blockings are apparently seen in the precipitation (Figure 14a) and root-zone soil moisture (Figure 14c) maps compared to runoff results (Figure 14b). Is it because low-res GCMs are spatially
interpolated into fine resolutions followed by statistical bias corrections? Why runoff results are smoother? Why not much spatial variability is seen in the PET plot (Figure 14d)? Furthermore, in previous spatial maps, the data-sparse regions are masked out. I suggest doing the same masking for this figure.

Yes, that is the reason the blockings are seen. The interpolation used was a "conservative
remapping" as opposed to a "bilinear interpolation", which may reduce some of the "blocking" in the visual representation of the results. We are currently investigating which interpolation method is more appropriate and how it may affect communication of the ensuing results.

Page 17 Line 26: You may want to show the same analysis for the historical period to be confident about the performance of GCM-driven hydrological projections in simulating the extreme events.

The GCMs do not represent a perfect historical analysis - they are set up with boundary conditions but because of parameterisations and lack of fine scale resolution, the GCMs are not a digital twin of Earth. Thus, the reference period of change needs to be the GCM historical period, as the actual historical period of observations or reanalysis has no bearing on the GCM simulated historical period. However, this is why we bias correct the output from the GCM, to bias correct for climate
variability, to make sure that the extremes fit within the distribution of the observations. In addition, (Vogel et al., 2023) has performed a thorough evaluation of the GCM variables and the GCM-driven hydrological projections, especially with regard to extremes (5-year maximum and 99, 99.5 and 99.9 percentiles).

**Editorial**

Page 1 Line 23: It is hard to understand this sentence without reading the main content. I suggest replacing 'one to output from a regional climate model forced by…' with 'one regional climate model (RCM) that is forced by …'

The suggested modification would not convey what was achieved, however, we do appreciate the difficulty in interpretation. We have modified it to now read (italicised words are additions):

"Three bias correction techniques were applied to *all* four CMIP5 global climate models (GCMs) and one *method* to a regional climate model (RCM) forced by the same four GCMs, resulting in a 16-member ensemble of bias-corrected GCM data for each emission scenario."

Careful proofreading is required throughout the article. Typical issues are:

- Missing commas: For example, in Page 3 Line 11-13, there should be a comma after 'To address these deficits in hydrological projections'.

  o    Modified as suggested.

- Sentences too long to read: For example, in Page 6 Line 1-4, it is better to break this
sentence into two or more before the words 'hence' and 'nevertheless'.

  o    Split into two sentences. It now reads:

  "Due to Australia's large size and geographical location, the climate of Australia varies markedly from the tropical north to the temperate south. As such, nationally-averaged precipitation may not provide meaningful insight from a climatological perspective,
nevertheless, it does impart interpretation of how our choice of GCMs occupies the phase space spanned by the CMIP5 models."

- Duplicate descriptions: For example, the sentence in Page 10 Line 19-21 is a duplicate description of Page 10 Line 25. The first sentence in Section 6 is also mentioned before.

  o    Page 10, line 25 (now page 10, line 28) has been modified to: "Steps (1) and (2) have
different implementations depending on whether the correction to be applied is additive or multiplicative."

Furthermore, we have carefully gone through the manuscript, to shorten lengthy sentences and pay careful attention to grammar. This has been aided using grammar checking software. We hope the reviewer finds the new version satisfactory.

**References:**

Azarnivand, A., Sharples, W., Bende-michl, U., Shokri, A. and Srikanthan, S.: Analysing the uncertainty of modelling hydrologic states of AWRA-L – understanding impacts from parameter uncertainty for the National Hydrological Projections., 2022.

Bosshard, T., Carambia, M., Goergen, K., Kotlarski, S., Krahe, P., Zappa, M. and Schär, C.: Quantifying uncertainty sources in an ensemble of hydrological climate-impact projections, Water Resour. Res., 49(3), 1523–1536, doi:10.1029/2011WR011533, 2013.

CSIRO and Bureau of Meteorology: Climate Change in Australia Projections for Australia's Natural Resource Management Regions: Technical Report., 2015.

Dobler, C., Hagemann, S., Wilby, R. L. and StÃtter, J.: Quantifying different sources of uncertainty in hydrological projections in an Alpine watershed, Hydrol. Earth Syst. Sci., 16(11), 4343–4360, doi:10.5194/HESS-16-4343-2012, 2012.

Frost, A. J., Ramchurn A. and Smith A.: The Australian Landscape Water Balance model (AWRA-L v6) Technical Description of the Australian Water Resources Assessment Landscape model version 6. [online] Available from: www.bom.gov.au/other/copyright.shtml (Accessed 1 June 2021), 2018.

Giuntoli, I., Vidal, J. P., Prudhomme, C. and Hannah, D. M.: Future hydrological extremes: The uncertainty from multiple global climate and global hydrological models, Earth Syst. Dyn., 6(1), 267–285, doi:10.5194/ESD-6-267-2015, 2015.

Grose, M. R., Narsey, S., Trancoso, R., Mackallah, C., Delage, F., Dowdy, A., Di Virgilio, G., Watterson, I., Dobrohotoff, P., Rashid, H. A., Rauniyar, S., Henley, B., Thatcher, M., Syktus, J., Abramowitz, G., Evans, J. P., Su, C. H. and Takbash, A.: A CMIP6-based multi-model downscaling ensemble to underpin climate change services in Australia, Clim. Serv., 30, 100368, doi:10.1016/J.CLISER.2023.100368, 2023.

Hope, P., Timbal, B. and Fawcett, R.: Associations between rainfall variability in the southwest and southeast of Australia and their evolution through time, Int. J. Climatol., 30(9), 1360–1371, doi:10.1002/joc.1964, 2010.

Joseph, J., Ghosh, S., Pathak, A. and Sahai, A. K.: Hydrologic impacts of climate change: Comparisons between hydrological parameter uncertainty and climate model uncertainty, J. Hydrol., 566, 1–22, 20   doi:10.1016/J.JHYDROL.2018.08.080, 2018.

Shepherd, T. G.: Bringing physical reasoning into statistical practice in climate-change science, Clim. Change, 169(1–2), doi:10.1007/S10584-021-03226-6, 2021.

Vogel, E., Johnson, F., Marshall, L., Bende-Michl, U., Wilson, L., Peter, J. R., Wasko, C., Srikanthan, S., Sharples, W., Dowdy, A., Hope, P., Khan, Z., Mehrotra, R., Sharma, A., Matic, V., Oke, A., Turner, M., 25   Thomas, S., Donnelly, C. and Duong, V. C.: Downscaling and bias correction for climate change impact studies, J. Hydrol. (in prep.), 1–41, 2022.

Vogel, E., Johnson, F., Marshall, L., Bende-Michl, U., Wilson, L., Peter, J. R., Wasko, C., Srikanthan, S., Sharples, W., Dowdy, A., Hope, P., Khan, Z., Mehrotra, R., Sharma, A., Matic, V., Oke, A., Turner, M., Thomas, S., Donnelly, C. and Duong, V. C.: An evaluation framework for downscaling and bias 30   correction in climate change impact studies, J. Hydrol., 622, 129693, doi:10.1016/J.JHYDROL.2023.129693, 2023.

---

## Author Comment (AC6)

Dear Professor Medlyn,

Thank you for this question. We didn't bias correct a humidity variable because it was not needed by the hydrological model (AWRA-L v6) that was used for the NHP.

5    We investigated calculating vapour pressure using the assumption that the dewpoint is tasmin and calculating vapour pressure using the FAO56 PM equation.

The results are displayed below. It is a bit hard to read the scale but the percentage difference plot ranges from -18 to 70 percent. This indicates that the assumption of using tasmin as the dewpoint is unrealistic (as expected).

Unfortunately, this means that you would need to apply your own bias correction to vapour pressure independently before using the NHP output for your
10   vegetation modelling.

As part of the Australian Climate Service (ACS – see https://www.acs.gov.au/), we will be undertaking bias correction of vapour pressure using CMIP6 data. Hopefully that data will useable for applications such as yours.

[Figure]

31-Dec-2019

Tmax

wind

Solar exposure

Tmin

Vapour pressure

fao56
12
0

Observed data

Fao56 (% difference)
70
-18

Vapour pressure (from Tmin)

Fao56 (using vp from Tmin)
11
0

$Vp = 6.108 \exp(17.27 * Tmin / (237.3 + Tmin)$